# Time Fairness in Online Knapsack Problems

**Adam Lechowicz**
University of Massachusetts Amherst
alechowicz@cs.umass.edu

**Rik Sengupta**
University of Massachusetts Amherst
rsengupta@cs.umass.edu

**Bo Sun**
University of Waterloo
bo.sun@uwaterloo.ca

**Shahin Kamali**
York University
kamalis@yorku.ca

**Mohammad Hajiesmaili**
University of Massachusetts Amherst
hajiesmaili@cs.umass.edu

## ABSTRACT

The online knapsack problem is a classic problem in the field of online algorithms. Its canonical version asks how to pack items of different values and weights arriving online into a capacity-limited knapsack so as to maximize the total value of the admitted items. Although optimal competitive algorithms are known for this problem, they may be fundamentally *unfair*, i.e., individual items may be treated inequitably in different ways. We formalize a practically-relevant notion of *time fairness* which effectively models a trade off between static and dynamic pricing in a motivating application such as cloud resource allocation, and show that existing algorithms perform poorly under this metric. We propose a parameterized deterministic algorithm where the parameter precisely captures the Pareto-optimal trade-off between fairness (static pricing) and competitiveness (dynamic pricing). We show that randomization is theoretically powerful enough to be simultaneously competitive and fair; however, it does not work well in experiments. To further improve the trade-off between fairness and competitiveness, we develop a nearly-optimal learning-augmented algorithm which is fair, consistent, and robust (competitive), showing substantial performance improvements in numerical experiments.

## 1 INTRODUCTION

The online knapsack problem (OKP) is a well-studied problem in online algorithms. It models a resource allocation process, in which one provider allocates a limited resource (i.e., the knapsack's *capacity*) to consumers (i.e., *items*) arriving sequentially in order to maximize the total return (i.e., optimally pack items subject to the capacity constraint). In OKP, as in many other online decision problems, there is a trade-off between *efficiency*, i.e., maximizing the value of packed items, and *fairness*, i.e., ensuring equitable treatment for different items across some desirable criteria.

**Example 1.1.** Consider a cloud computing resource accepting heterogeneous jobs online from clients sequentially. Each job includes a bid that the client is willing to pay, and an amount of resources required to execute it. The resource is not sufficiently large to service all of the incoming requests. We define the *quality* of a job as the ratio of the price paid by the client over the resources required.

How do we algorithmically solve the problem posed in Example 1.1? Note that the limited resource implies that the problem of accepting and rejecting items reduces precisely to OKP. If we cared only about the overall quality of accepted jobs, we would intuitively be solving the unconstrained online knapsack problem. However, it might be desirable for an algorithm to apply the same *quality criteria* to each job that arrives. As we will show in §2, existing optimal algorithms for OKP do not fulfill this second requirement. In particular, although two jobs may have *a priori* identical quality, the optimal algorithm discriminates between them based on their arrival time in the online queue: a typical job, therefore, has a higher chance of being accepted if it happens to arrive earlier rather than later. Preventing these kinds of choices while still maintaining competitive standards of overall quality will form this work's focus. We briefly note that solving OKP is equivalent to designing a *pricing strategy* – for example, the minimum value that the online knapsack will accept can be interpreted as a *posted price* (Zhang et al., 2017). The notion of fairness discussed above

formalizes a design goal of *static (flat-rate) pricing*, which is often more desirable from an end-user perspective, mainly due to its simplicity. A customer can price their bid at the static posted price, having confidence that their job will be accepted. This is in contrast with existing optimal algorithms for OKP, which correspond to *dynamic (variable) pricing* strategies – these are efficient in terms of revenue maximization (Al-Roomi et al., 2013; Chakraborty et al., 2013; Borenstein et al., 2002), but undesirable from an end-user perspective due to volatility, uncertainty, and inequity. Of note for this work, the ridesharing industry successfully uses a hybrid model of *surge pricing* (Hall et al., 2015).

**Precursors to this work.** OKP was first studied by Marchetti-Spaccamela and Vercellis (1995), with important optimal results following in Zhou et al. (2008). Recent work (Im et al., 2021; Zeynali et al., 2021; Böckenhauer et al., 2014) has explored this problem with advice and/or learning augmentation, which we also consider – in particular, we consider a prediction model which is similar to the frequency predictions explored by Im et al. (2021). In the context of fairness, our work is most closely aligned with the notions of time fairness recently studied for prophet inequalities (Arsenis and Kleinberg, 2022) and the secretary problem (Buchbinder et al., 2009), which considered stochastic and random order input arrivals. To the best of our knowledge, our work is the first to consider *any* notions of fairness in OKP while considering adversarial inputs. We present a comprehensive review of related studies on the knapsack problem and fairness in Appendix A.

**Our contributions.** First, we introduce a natural notion of time-independent fairness. We show (Thm. 3.3) that the original notion (Arsenis and Kleinberg, 2022) is *too restrictive* for OKP, and motivate a revised definition which is feasible. We design a deterministic algorithm to achieve the desired fairness properties and show that it captures the Pareto-optimal trade-off between fairness and competitiveness (Thms. 4.5, 4.6). We further investigate randomization, showing that a randomized algorithm can achieve optimal competitiveness and fairness in theory (Thm. 4.7, Prop. 4.8); however, in trace-driven experiments, this underperforms significantly. Motivated by this observation, we incorporate predictions to *simultaneously* improve fairness and competitiveness. We introduce a fair, deterministic learning-augmented algorithm with bounded consistency and robustness (Thms. 4.11, 4.12), and we show that improving this significantly is essentially impossible (Thm. 4.13). Finally, we present numerical experiments evaluating our algorithms.

Our research has three primary technical contributions. First, our definition of $\alpha$-conditional time-independent fairness, which is generalizable to many online problems, captures a natural perspective on fairness in online settings, particularly when the impacts of static or dynamic pricing are of interest. Second, we present a constructive lower-bound technique to derive a Pareto-optimal trade-off between fairness and competitiveness, illustrating the inherent challenges in this problem. This provides a strong understanding of the necessary compromise between fair decision-making and efficiency in OKP, and gives insight into the results achievable for other online problems. Lastly, we design a nearly-optimal learning-augmented algorithm which matches existing results for learning-augmented OKP while introducing fairness guarantees, showing that predictions can simultaneously improve an online algorithm's fairness and performance.

## 2 PROBLEM & PRELIMINARIES

**Problem formulation.** In the online knapsack problem (OKP), we have a knapsack (resource) with capacity $B$ and items arriving online. We denote an instance $\mathcal{I}$ of OKP as a multiset of $n$ items, where each item has a *value* and a *weight*. Formally, $\mathcal{I} = \left[ (v_j, w_j)_{j=1}^n \right]$. We assume that $v_j$ and $w_j$ correspond to the value and weight respectively of the $j$th item in the arrival sequence, where $j$ also denotes the arrival time of this item. We denote by $\Omega$ the (infinite) set of all feasible input instances.

The objective in OKP is to accept items into the knapsack to maximize the sum of values while not violating the capacity limit of $B$. As is standard in the literature on online algorithms, at each time step $j$, the algorithm is presented with an item, and must immediately decide whether to accept it ($x_j = 1$) or reject it ($x_j = 0$). The offline version of OKP (i.e., Knapsack) is a classical combinatorial problem with strong connections to resource allocation. It can be summarized as $\max \sum_{j=1}^n v_j x_j$, s.t. $\sum_{j=1}^n w_j x_j \leq B$, $x_j \in \{0, 1\}$ for all $j \in [n]$.

Knapsack is a canonical NP-complete problem (strongly NP-complete for non-integral inputs (Wojtczak, 2018)). There is a folklore pseudopolynomial algorithm known for the exact problem. Knapsack is one of the hardest but most ubiquitous problems arising in applications today.

**Competitive analysis.** OKP has been extensively studied under the framework of *competitive analysis*, where the goal is to design an online algorithm that maintains a small *competitive ratio* (Borodin et al., 1992), i.e., performs nearly as well as the offline optimal. For online algorithm ALG and offline algorithm OPT, the competitive ratio for a maximization problem is defined as $\max_{\mathcal{I} \in \Omega} \mathsf{OPT}(\mathcal{I})/\mathsf{ALG}(\mathcal{I})$, where $\mathsf{OPT}(\mathcal{I})$ is the optimal profit on input $\mathcal{I}$, and $\mathsf{ALG}(\mathcal{I})$ is the profit obtained by the online algorithm on the same. This ratio is always at least one, and lower is better.

**Assumptions and additional notation.** We assume that the set of *value densities* $\{v_j/w_j\}_{j \in [n]}$ has bounded support, i.e., $v_j/w_j \in [L, U]$ for all $j$, where $L$ and $U$ are known. These are standard assumptions in the literature for many online problems, including OKP, one-way trading, and online search; without them, the competitive ratio of any online algorithm is unbounded (Marchetti-Spaccamela and Vercellis, 1995; Zhou et al., 2008). We also adopt an assumption from the literature on OKP, assuming that individual item weights are sufficiently small compared to the knapsack's capacity (Zhou et al., 2008). This is reasonable in practice and necessary for a meaningful result. For the rest of the paper, we will assume WLOG that $B = 1$. We can scale everything down by a factor of $B$ otherwise. Let $z_j \in [0, 1]$ denote the *knapsack utilization* when the $j$th item arrives, i.e. the fraction of the knapsack's total capacity that is currently full.

**Existing results.** Prior work on OKP has resulted in an optimal deterministic algorithm for the problem described above, shown by Zhou et al. (2008) in a seminal work using the framework of online threshold-based algorithms (OTA). In OTA, a carefully designed *threshold function* is used to make decisions at each time step. This threshold is specifically designed such that greedily accepting inputs whose values meet or exceed the threshold at each step provides competitive guarantees. The OTA framework has seen success in the related online search and one-way trading problems (Lee et al., 2022; Sun et al., 2021; Lorenz et al., 2008) as well as OKP (Sun et al., 2022; Yang et al., 2021). The algorithm shown by Zhou et al. (2008) is a deterministic threshold-based algorithm that achieves a competitive ratio of $\ln(U/L) + 1$; they also show that this is the optimal competitive ratio for any deterministic *or* randomized algorithm. We henceforth refer to this algorithm as the ZCL algorithm. In the ZCL algorithm, items are admitted based on a monotonically increasing threshold function $\Phi(z) = (Ue/L)^z (L/e)$, where $z \in [0, 1]$ is the current utilization (see Fig. **1(a)**). The $j$th item in the sequence is accepted if it satisfies $v_j/w_j \geq \Phi(z_j)$, where $z_j$ is the utilization at the time of the item's arrival. This algorithm achieves the optimal competitive ratio (Zhou et al., 2008, Thms. 3.2, 3.3).

## 3 TIME FAIRNESS

In Example 1.1, the infringed constraint was one of *time fairness*. In this section, inspired by Arsenis and Kleinberg (2022) who explore the concept of time fairness in the context of prophet inequalities, we formally define the notion in the context of OKP. We relegate formal proofs to Appendix B.

### 3.1 TIME-INDEPENDENT FAIRNESS (TIF)

In Example 1.1, it is reasonable to ask that the probability of an item's admission into the knapsack depend solely on its value density $x$, and not on its arrival time $j$. This is a natural generalization to OKP of the *time-independent fairness* constraint, proposed by Arsenis and Kleinberg (2022).

**Definition 3.1** (Time-Independent Fairness (TIF) for OKP)**.** *An* OKP *algorithm* ALG *is said to satisfy* TIF *if there exists a function* $p : [L, U] \to [0, 1]$ *such that:*

$$\Pr[\mathsf{ALG} \textit{ accepts } j\textit{th item in } \mathcal{I} \mid v_j/w_j = x] = p(x), \quad \textit{for all } \mathcal{I} \in \Omega, j \in [n], x \in [L, U].$$

In other words, the probability of admitting an item of value density $x$ depends only on $x$, and not on its arrival time. Note that this definition makes sense only in the online setting. We start by noting that the ZCL algorithm is not TIF.

**Observation 3.2.** *The* ZCL *algorithm (Zhou et al., 2008) is not* TIF.

We seek to design an algorithm for OKP that satisfies TIF while being competitive against an optimal offline solution. Given Observation 3.2, a natural question is whether any such algorithm exists that maintains a bounded competitive ratio, perhaps in the presence of some additional information about the input. For instance, we could seek to leverage ML predictions in the spirit of Im et al. (2021), who present the first work, to our knowledge, that incorporates ML predictions into OKP. They use

*frequency predictions*, which give upper and lower bounds on the total weight of items with a given value density in the input instance. Formally, if $s_x := \sum_{j:v_j/w_j=x} w_j$ is the total weight of items with value density $x$, we get a predicted pair $(\ell_x, u_x)$ satisfying $\ell_x \leq s_x \leq u_x$ for all $x \in [L, U]$. The OKP instance is assumed to respect these predictions, although it can be adversarial within these bounds. It is conceivable that additional information such as the total number of items or these frequency predictions could enable a nontrivial OKP algorithm to guarantee TIF.

Of course, the *trivial* algorithm which rejects all items is TIF, as the probability of accepting any item is zero. We now show that there is no other algorithm for OKP that achieves our desired conditions simultaneously, even for perfect predictions, i.e., $\ell_x = u_x = s_x$ for all $x \in [L, U]$ (i.e., the algorithm knows each $s_x$ in advance), and even with advance knowledge of the length of the input sequence.

**Theorem 3.3.** *There is no nontrivial algorithm for* OKP *that guarantees* TIF *without additional information about the input. Further, even if the input length $n$ or perfect frequency predictions as defined in Im et al. (2021) are known in advance, no nontrivial algorithm can guarantee* TIF.

Theorem 3.3 shows that TIF can be essentially closed off as a candidate for a fairness constraint on competitive algorithms for OKP, even in the presence of reasonable information. An algorithm that knows both $n$ and the weights of input items is closer in spirit to an offline algorithm than an online one. We remark that (Arsenis and Kleinberg, 2022, Thm. 6.2) shows a similar impossibility result, wherein certain secretary problem variants which must hire at least one candidate cannot satisfy TIF.

## 3.2 CONDITIONAL TIME-INDEPENDENT FAIRNESS (CTIF)

Motivated by the results in §3.1, we now present a revised but still natural notion of fairness in Definition 3.4. This notion relaxes the constraint and narrows the scope of fairness to consider items which arrive while the knapsack's utilization is within a subinterval of the knapsack's capacity.

**Definition 3.4** ($\alpha$-Conditional Time-Independent Fairness ($\alpha$-CTIF) for OKP). *For $\alpha \in [0, 1]$, an* OKP *algorithm* ALG *is said to satisfy $\alpha$-CTIF if there exists a subinterval $\mathcal{A} = [a, b] \subseteq [0, 1]$ where $b - a = \alpha$, and a function $p : [L, U] \to [0, 1]$ such that:*

$$\Pr\left[\text{ALG } accepts \text{ } jth \text{ } item \text{ } in \text{ } \mathcal{I} \mid (v_j/w_j = x) \text{ } and \text{ } (z_j + w_j \in \mathcal{A})\right] = p(x),$$

$$for \text{ } all \text{ } \mathcal{I} \in \Omega, j \in [n], x \in [L, U].$$

Note that if $\alpha = 1$, then $\mathcal{A} = [0, 1]$, and any item that arrives while the knapsack still has the capacity to admit it is considered within this definition. (i.e., 1-CTIF is the same as TIF provided that the knapsack has capacity remaining for the item under consideration). Furthermore, the larger the value of $\alpha$, the stronger the fairness guarantee, but even the strongest 1-CTIF is strictly weaker than TIF. This notion circumvents the challenges of TIF and is feasible in the online setting while preserving competitive guarantees. However, we note that the canonical ZCL algorithm still is not 1-CTIF.

**Observation 3.5.** *The* ZCL *algorithm (Zhou et al., 2008) is not* 1-CTIF.

In the context of Example 1.1, $\alpha$-CTIF formalizes a *trade-off* between static (flat-rate) and dynamic (variable) pricing schemes. For instance, a cloud resource provider could generally provide a simple, interpretable acceptance threshold for their customers (static pricing within the fair subinterval), then switch to a dynamic pricing strategy when the resource is highly utilized or under utilized in order to maximize revenue. For this motivating application, the tunable value of $\alpha$ is a benefit to the resource provider, who can modulate the "fairness parameter" up or down to attract customers or manage high demand, respectively. A real-world example of such a transition between static and dynamic pricing can be found in the surge pricing model used successfully by many ride sharing platforms (Castillo et al., 2017). In cloud applications, dynamic pricing defines the price of Virtual Machines (VMs) based on electricity price or demand, in contrast to the more common model of static pricing (Alzhouri et al., 2017; Zhang et al., 2020). Our definition of $\alpha$-CTIF also offers a fairness concept that is both achievable for OKP and potentially adaptable to other online problems, such as online search (Lorenz et al., 2008), one-way trading (El-Yaniv et al., 2001), bin-packing (Balogh et al., 2017; Johnson et al., 1974), and single-leg revenue management (Ma et al., 2021; Balseiro et al., 2023). We are now ready to present our main results, including deterministic and learning-augmented algorithms which satisfy $\alpha$-CTIF and provide competitive guarantees.

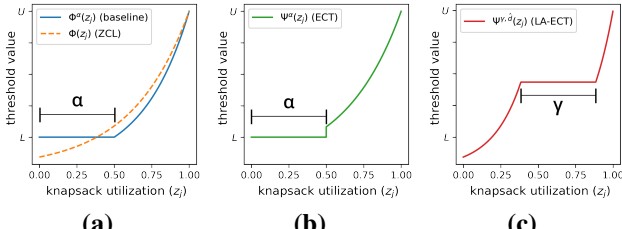 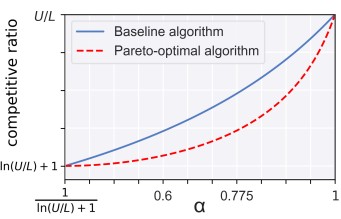

Figure 1: Plotting the threshold functions for several OKP algorithms. **(a)** ZCL (§2) and the baseline algorithm (see §3); **(b)** ECT (§4.1); **(c)** LA-ECT (§4.3)

Figure 2: Trade-off for the baseline algorithm and the Pareto-optimal lower bound. $U/L = 5$.

## 4 ONLINE FAIR ALGORITHMS

In this section, we start with some simple fairness-competitiveness trade-offs. In §4.1, we develop competitive algorithms which satisfy CTIF constraints. Finally, we explore the power of randomization in §4.2 and predictions in §4.3. For the sake of brevity, we relegate most proofs to Appendix C, but provide proof sketches in a few places. We start with a warm-up result capturing inherent trade-offs through an examination of constant threshold-based algorithms for OKP. Such an algorithm sets a single threshold $\phi$ and greedily accepts any items with value density $\geq \phi$.

**Proposition 4.1.** *Any constant threshold-based algorithm for* OKP *is* 1*-CTIF. Furthermore, any constant threshold-based* deterministic *algorithm for* OKP *cannot be better than* $(U/L)$*-competitive.*

How does this trade-off manifest itself in the ZCL algorithm? We know from Observation 3.5 that ZCL is not 1-CTIF. In the next result, we show that this algorithm *is*, in fact, $\alpha$-CTIF for some $\alpha > 0$.

**Proposition 4.2.** *The* ZCL *algorithm is* $\frac{1}{\ln(U/L)+1}$*-CTIF.*

### 4.1 PARETO-OPTIMAL DETERMINISTIC ALGORITHMS

We present $\alpha$-CTIF threshold-based deterministic algorithms which maintain competitive bounds in terms of $U$, $L$, and $\alpha$. We'll start with a simple baseline idea to contextualize our later results.

**Baseline algorithm.** Proposition 4.2 together with the competitive optimality of the ZCL algorithm shows that we can get a competitive, $\alpha$-CTIF algorithm for OKP when $\alpha \leq 1/\ln(U/L)+1$. Now, let $\alpha \in [1/\ln(U/L)+1, 1]$ be a parameter capturing our desired fairness. To design a competitive $\alpha$-CTIF algorithm, we can use Proposition 4.1 and apply it to the "constant threshold" portion of the utilization interval, as described in the proof of Proposition 4.2. The idea is to define a new threshold function that "stretches" out the portion of the threshold from $[0, 1/\ln(U/L)+1]$ (where $\Phi(z) \leq L$) to our desired length of a subinterval. The intuitive idea above is captured by function $\Phi^\alpha(z) = (Ue/L)^{z-\ell/1-\ell} (L/e)$, where $\ell = \alpha + \alpha-1/\ln(U/L)$ (Fig. **1(a)**). Note that $\Phi^\alpha(z) \leq L$ for all $z \leq \alpha$.

**Theorem 4.3.** *For* $\alpha \in [1/\ln(U/L)+1, 1]$*, the baseline algorithm is* $\frac{U[\ln(U/L)+1]}{L\alpha[\ln(U/L)+1]+(U-L)(1-\ell)}$*-competitive and $\alpha$-CTIF for* OKP.

The proof (Appendix C) relies on keeping track of the common items picked by the algorithm and an optimal offline one, and approximating the total value obtained by the algorithm by an integral. Although this algorithm is competitive and $\alpha$-CTIF, in the following, we will demonstrate a *gap* between the algorithm and the Pareto-optimal lower bound from Theorem 4.5 (Fig. 2).

**Lower bound.** We consider how the $\alpha$-CTIF constraint impacts the achievable competitive ratio for *any* online deterministic algorithm. To show such a lower bound, we first construct a family of special instances and then show that for any $\alpha$-CTIF online deterministic algorithm (which is not necessarily threshold-based), the competitive ratio is lower bounded under the constructed special instances. It is known that difficult instances for OKP occur when items arrive at the algorithm in a non-decreasing order of value density (Zhou et al., 2008; Sun et al., 2020). We now formalize such a family of instances $\{\mathcal{I}_x\}_{x\in[L,U]}$, where $\mathcal{I}_x$ is called an $x$-continuously non-decreasing instance.

**Definition 4.4.** *Let* $N, m \in \mathbb{N}$ *be sufficiently large, and* $\delta := (U - L)/N$*. For* $x \in [L, U]$*, an instance* $\mathcal{I}_x \in \Omega$ *is $x$-continuously non-decreasing if it consists of* $N_x := \lceil (x - L)/\delta \rceil + 1$ *batches of items and the $i$-th batch* $(i \in [N_x])$ *contains $m$ items with value density $L + (i - 1)\delta$ and weight $1/m$.*

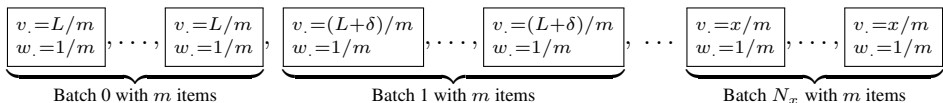

Figure 3: $\mathcal{I}_x$ consists of $N_x$ batches of items, arriving in increasing order of value density.

Note that $\mathcal{I}_L$ is simply a stream of $m$ items, each with weight $^1/m$ and value density $L$. See Fig. 3 for an illustration of an $x$-continuously non-decreasing instance.

**Theorem 4.5.** *No $\alpha$-CTIF deterministic online algorithm for* OKP *can achieve a competitive ratio smaller than* $\frac{W\left(\frac{U(1-\alpha)}{L\alpha}\right)}{1-\alpha}$, *where* $W(\cdot)$ *is the Lambert W function.*

*Proof sketch.* Consider the "fair" utilization region of length $\alpha$ for any $\alpha$-CTIF algorithm, and consider the lowest value density $v$ that it accepts in this interval. We show (Lemma C.1) that it is sufficient to focus on $v = L$ since the competitive ratio is strictly worst if $v > L$. Under an instance $\mathcal{I}_x$ (for which the offline optimum is $x$), any $\beta'$-competitive deterministic algorithm obtains a value of $\alpha L + \int_{\alpha\beta'L}^{x} u\,dg(u)$, where the integrand represents the approximate value obtained from items with value density $u$ and weight allocation $dg(u)$. Using Grönwall's Inequality yields a necessary condition for the competitive ratio $\beta'$ to be $\ln(U/\alpha\beta'L)/\beta' = 1 - \alpha$, and the result follows. $\square$

Motivated by this Pareto-optimal trade-off, in the following, we design an improved algorithm that closes the theoretical gap between the intuitive baseline algorithm and the lower bound by developing a new threshold function utilizing a discontinuity to be more selective outside the fair region.

**Extended Constant Threshold (ECT) for** OKP**.** Let $\alpha \in [^1/\ln(U/L)+1, 1]$ be a fairness parameter. Given this $\alpha$, we define a threshold function $\Psi^\alpha(z)$ on the interval $z \in [0, 1]$, where $z$ is the current knapsack utilization. $\Psi^\alpha$ is defined as follows (Fig. **1(b)**):

$\Psi^\alpha(z) = L$ for $z \in [0, \alpha]$, and $\Psi^\alpha(z) = Ue^{\beta(z-1)}$ for $z \in (\alpha, 1]$, where $\beta = \frac{W\left(\frac{U(1-\alpha)}{L\alpha}\right)}{1-\alpha}$.

Let us call this algorithm ECT[$\alpha$]. The following captures its fairness/competitiveness trade-off.

**Theorem 4.6.** *For any $\alpha \in [^1/\ln(U/L)+1, 1]$,* ECT[$\alpha$] *is $\beta$-competitive and $\alpha$-CTIF.*

*Proof sketch.* For an instance $\mathcal{I} \in \Omega$, suppose ECT[$\alpha$] terminates with the utilization at $z_T$, and let $W$ and $V$ denote the total weight and total value of the common items picked by ECT[$\alpha$]($\mathcal{I}$) and OPT($\mathcal{I}$) respectively. Using OPT($\mathcal{I}$) $\leq V + \Psi^\alpha(z_T)(1 - W)$, we can bound the ratio OPT($\mathcal{I}$)/ECT[$\alpha$]($\mathcal{I}$) by $\Psi^\alpha(z_T)/\sum_{j \in \mathcal{P}} \Psi^\alpha(z_j)w_j$, where $\mathcal{P}$ is the set of items picked by ECT[$\alpha$]. Taking item weights to be very small, we can approximate this denominator by $\int_0^{z_T} \Psi^\alpha(z)dz$. This quantity can be lower bounded by $L\alpha$ when $z_T = \alpha$, and by $\Psi^\alpha(z_T)/\beta$ when $z_T > \alpha$, using some careful case analysis. In either case, we can bound the competitive ratio by $\beta$. The $\alpha$-CTIF result is by definition. $\square$

## 4.2 RANDOMIZATION HELPS

In a follow-up to Theorem 4.6, we can ask whether a randomized algorithm for OKP exhibits similar trade-offs as in the deterministic setting. We refute this, showing that randomization *can* satisfy 1-CTIF while simultaneously obtaining the optimal competitive ratio in expectation.

Motivated by related work on randomized OKP algorithms, as well as by Theorem 4.1, we highlight one such randomized algorithm and show that it provides the best possible fairness guarantee. In addition to their optimal deterministic ZCL algorithm, Zhou et al. (2008) also show a related randomized algorithm for OKP; we will henceforth refer to this algorithm as ZCL-Randomized. ZCL-Randomized samples a constant threshold $\phi$ from the continuous distribution $\mathcal{D}$ over the interval $[0, U]$, with probability density function $f(x) = c/x$ for $L \leq x \leq U$, and $f(x) = c/L$ for $0 \leq x \leq L$, where $c = 1/[\ln(U/L) + 1]$. When item $j$ arrives, ZCL-Randomized accepts it iff $v_j/w_j \geq \phi$ and $z_j + w_j \leq 1$ (i.e., the item's value density is above the threshold, and there is enough space for it). The following result shows the competitive optimality of ZCL-Randomized.

**Theorem 4.7** (Zhou et al. (2008), Thms. 3.1, 3.3)**.** ZCL-Randomized *is $\ln(U/L) + 1$-competitive over every input sequence. Furthermore, no online algorithm can achieve a better competitive ratio.*

ZCL-Randomized is trivially 1-CTIF by Proposition 4.1. We state this as a proposition without proof.

**Proposition 4.8.** ZCL-Randomized *is* 1-CTIF.

Although ZCL-Randomized seems to provide the "best of both worlds" from a theoretical perspective, in practice (see §5) we find that it falls short compared to the deterministic algorithms. We believe this is consistent with empirical results for caching (Reineke, 2014), where randomized techniques are theoretically superior but not used in practice. Motivated by this and our deterministic lower bounds, in the next section we draw inspiration from the emerging field of learning-augmented algorithms.

### 4.3 PREDICTION HELPS

In this section, we explore how predictions might help to achieve a better trade-off between competitiveness and fairness. We propose an algorithm, LA-ECT, which integrates predictions and matches prior consistency-robustness bounds for learning-augmented OKP, while introducing CTIF guarantees. We give a corresponding lower bound for the fairness-consistency-robustness trade-off which shows LA-ECT is *nearly-optimal*. To start, we introduce and formalize our prediction model.

**Prediction model.** Consider a 1-CTIF constant threshold algorithm as highlighted in Proposition 4.1. Although Proposition 4.1 indicates that any such an algorithm with threshold $\phi > L$ cannot have a nontrivial competitive ratio; we know intuitively that increasing $\phi$ makes the algorithm more selective in admitting items. The question, then, is *what constant threshold minimizes the competitive ratio on a typical instance*? We build on prior work (Im et al., 2021) considering frequency predictions, which give upper and lower bounds of value densities amongst items in a typical sequence. We show that similar information about the optimal offline solution allows us to recover and leverage a critical threshold value $d_\gamma^\star$, which we define as follows.

**Definition 4.9** (Critical threshold $d_\gamma^\star$). *Consider function* $\rho_\mathcal{I}(x) : [L, U] \to [0, 1]$. $\rho_\mathcal{I}(x)$ *describes the fraction of the offline optimal solution's total value contributed by items whose value density* $\geq x$ *on a sequence* $\mathcal{I} \in \Omega$. *Define* $d_\gamma^\star$ *as the largest $x$ satisfying* $\gamma/2 \leq \rho_\mathcal{I}(x)$ *for a given* $\gamma \in [0, 1]$.

It is reasonable to assume that $\rho_\mathcal{I}(x)$ can be learned for typical instances from historical data, since a model can calculate the hindsight optimal solutions and learn the frequencies of packed items. Such predictions have been shown to be PAC learnable by (Canonne, 2020).
Let $\hat{\rho}_\mathcal{I}^{-1}(y) : y \in [0, 1] \to [L, U]$ denote a black-box predictor, where $\hat{\rho}_\mathcal{I}^{-1}(y)$ is the predicted inverse function of $\rho_\mathcal{I}(x)$. By a single query $\gamma/2$ made to the predictor, we can obtain $\hat{d}_\gamma = \rho_\mathcal{I}^{-1}(\gamma/2)$, which predicts the critical value $d_\gamma^\star$. Using this prediction model, we present an online algorithm that incorporates the prediction into its threshold function. We follow the emerging literature on learning-augmented algorithms, where algorithms are evaluated using *consistency* and *robustness* (Lykouris and Vassilvtiskii, 2018; Purohit et al., 2018). Consistency is defined as the competitive ratio of the algorithm when predictions are accurate, while robustness is the worst-case competitive ratio over any prediction errors. These metrics jointly measure how an algorithm exploits accurate predictions and ensures bounded competitiveness with poor predictions.

**Learning-Augmented Extended Constant Threshold (LA-ECT) for OKP.** Fix a *confidence parameter* $\gamma \in [0, 1]$ ($\gamma = 0$ and $\gamma = 1$ correspond to *untrusted* and *fully trusted* predictions respectively), and $\rho_\mathcal{I}^{-1}(\cdot)$ is the black-box predictor. We define a threshold function $\Psi^{\gamma, \hat{d}}(z)$ on the utilization interval $z \in [0, 1]$ as follows (see Fig. **1(c)**):

$$\Psi^{\gamma, \hat{d}}(z) = \begin{cases} (Ue/L)^{\frac{z}{1-\gamma}} (L/e) & z \in [0, \kappa], \\ \hat{d}_\gamma := \rho_\mathcal{I}^{-1}(\gamma/2) & z \in (\kappa, \kappa + \gamma), \\ (Ue/L)^{\frac{z-\gamma}{1-\gamma}} (L/e) & z \in [\kappa + \gamma, 1], \end{cases} \tag{1}$$

where $\kappa \in [0, 1]$ is the point where $(Ue/L)^{(z/1-\gamma)}(L/e) = \hat{d}_\gamma$. Note that when $\gamma \to 1$, $\kappa \to 0$ and the resulting threshold function is constant at $\hat{d}_\gamma$. Call the resulting threshold-based algorithm LA-ECT[$\gamma$]. The following results characterize the fairness of this algorithm, followed by the results for consistency and robustness, respectively. We omit the proof for Proposition 4.10.

**Proposition 4.10.** LA-ECT[$\gamma$] *is* $\gamma$-CTIF.

**Theorem 4.11.** *For any* $\gamma \in (0, 1]$, *and any* $\mathcal{I} \in \Omega$, LA-ECT[$\gamma$] *is* $\frac{2}{\gamma}$-*consistent*.

*Proof sketch.* Assuming the black-box predictor is accurate, the consistency of the algorithm can be analyzed against a semi-online algorithm called ORACLE$_\gamma^\star$ (Lemma C.2), which we show to be

within a constant factor of OPT. The idea is to compare the utilization and the total value attained for $\mathcal{I}$ between $\mathsf{ORACLE}_\gamma^\star$ and $\mathsf{LA\text{-}ECT}[\gamma]$ carefully. In a case analysis, we can show that $\mathsf{LA\text{-}ECT}[\gamma]$ accepts every item accepted by $\mathsf{ORACLE}_\gamma^\star$, thus inheriting the same competitive upper bound. $\qquad\square$

**Theorem 4.12.** *For any $\gamma \in [0,1]$, and any $\mathcal{I} \in \Omega$, $\mathsf{LA\text{-}ECT}[\gamma]$ is $\frac{1}{1-\gamma}\left(\ln(U/L)+1\right)$-robust.*

*Proof sketch.* As in the proof of Theorem 4.6, we can bound $\mathsf{OPT}(\mathcal{I})/\mathsf{LA\text{-}ECT}[\gamma](\mathcal{I})$ for any $\mathcal{I} \in \Omega$ by $\Psi^{\gamma,\hat{d}}(z_T)/\sum_{j \in \mathcal{P}} \Psi^{\gamma,\hat{d}}(z_j)w_j$, where the notation follows from before. By assuming that the individual weights are much smaller than 1, we can approximate this denominator by an integral once again, and consider three cases. When $z_T < \kappa$, we can bound the integral by $(1-\gamma)\Psi^{\gamma,\hat{d}}(z_T)/\ln(Ue/L)$, which suffices for our bound. When $\kappa \leq z_T < \kappa+\gamma$, we can show an improvement from the previous case, inheriting the same worst-case bound. Finally, when $z_T \geq \kappa+\gamma$, we can inherit the same approximation as in the first case with a negligible additive term of $\hat{d}\gamma$. $\qquad\square$

It is reasonable to ask whether improvements to Theorems 4.11 or 4.12 are possible, while still maintaining fairness guarantees. We now show that the consistency and robustness of $\mathsf{LA\text{-}ECT}[\gamma \to 1]$ is nearly optimal. We relegate the proof to the appendix.

**Theorem 4.13.** *For any learning-augmented online algorithm $\mathsf{ALG}$ which satisfies 1-CTIF, one of the following holds: Either $\mathsf{ALG}$'s consistency is $> 2\sqrt{U/L}-1$, or $\mathsf{ALG}$ has unbounded robustness. Furthermore, the consistency of any algorithm is lower bounded by $2-\varepsilon^2/1+\varepsilon$, where $\varepsilon = \sqrt{L/U}$.*

## 5 NUMERICAL EXPERIMENTS

In this section, we present numerical experiments for OKP algorithms in the context of the online job scheduling problem from Example 1.1. We evaluate our proposed algorithms, including ECT and LA-ECT, against existing algorithms from the literature.[1]

**Setup.** To validate the performance of our algorithms and quantify the empirical trade-off between fairness and efficiency (resp. static and dynamic pricing), we conduct experiments on synthetic instances for OKP, which emulate a typical online cloud allocation task. We simulate different value density ratios $U/L \in \{100, 500, 2500\}$ by setting $L$ and $U$ accordingly. We generate value densities for each item in the range $[U, L]$ according to a power-law distribution (giving relatively few jobs with very high value, and many items with average and low values). We consider a one-dimensional knapsack (i.e., server w/ single CPU) and set the capacity to 1. Weights are assigned uniformly randomly and are small compared to the total knapsack capacity (e.g., the maximum weight is 0.05). We report the cumulative density functions of the empirical competitive ratios, which show the average and the worst-case performances of several algorithms as described below.

**Comparison algorithms.** As a benchmark, we calculate the optimal offline solution for each instance, allowing us to report the empirical competitive ratio for each tested algorithm. In the setting *without predictions*, we compare our proposed ECT algorithm against three others: ZCL, the baseline $\alpha$-CTIF algorithm, and ZCL-Randomized (§2, §4.1, and §4.2 respectively). For ECT, we set several values for $\alpha$ to show how performance degrades with more stringent fairness guarantees. In the setting *with predictions*, we compare our proposed LA-ECT algorithm (§4.3) against two other algorithms: ZCL and ECT. Simulated predictions are obtained by first solving for the actual value $d_\gamma^\star$ (Defn. 4.9). For simulated *prediction error*, we set $\hat{d}_\gamma = d_\gamma^\star(1+\eta)$, where $\eta \sim \mathcal{N}(0, \sigma)$. In this setting, we fix a single value for $U/L$ and report results for different levels of error obtained by changing $\sigma$.

**Experimental results.** We report the empirical competitive ratio, and intuitively we expect *worse* empirical competitiveness for algorithms which provide stronger fairness guarantees. In the first experiment, we test algorithms for OKP in the setting *without predictions*, for several values of $U/L$. In Fig. 4, we show the CDF of the empirical competitive ratios for each tested value of $U/L$. We observe that the performance of ECT[$\alpha$] exactly reflects the fairness parameter $\alpha$, meaning that a greater value of $\alpha$ corresponds with a worse competitive ratio, as shown in Theorems 4.5 and 4.6. Reflecting the theoretical results, ECT[$\alpha$] outperforms the baseline algorithm for $\alpha = 0.66$ by an average of $20.9\%$ across all experiments. Importantly, we also observe that ZCL-Randomized performs poorly

---

[1]Our code is available at `https://github.com/adamlechowicz/fair-knapsack`.

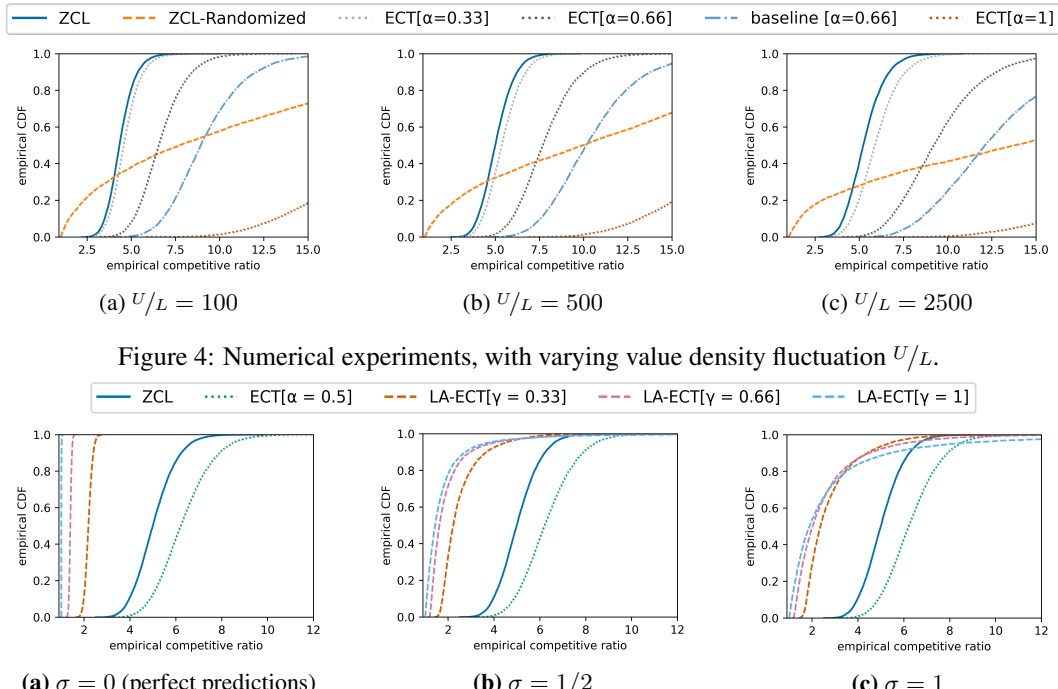

(a) $U/L = 100$

(b) $U/L = 500$

(c) $U/L = 2500$

Figure 4: Numerical experiments, with varying value density fluctuation $U/L$.

(a) $\sigma = 0$ (perfect predictions)

(b) $\sigma = 1/2$

(c) $\sigma = 1$

Figure 5: Learning-augmented experiments, with $U/L = 500$ and different prediction errors.

compared to the deterministic algorithms. This is a departure from the theory since Theorem 4.7 and Proposition 4.8 dictate that ZCL-Randomized should be optimally competitive and completely fair. We attribute this gap to the "one-shot" randomization used in the design of ZCL-Randomized – although picking a single random threshold yields good performance in expectation, the probability of picking a *bad* threshold is high. Coupled with the observation that ZCL and ECT often significantly outperform their theoretical bounds, this leaves ZCL-Randomized at a disadvantage.

In the second experiment, we investigate the impact of prediction error in the setting *with predictions*. We fix $U/L = 500$ and vary $\sigma$, which is the standard deviation of the multiplicative error $\eta$. In Fig. 5, we show the CDF of the empirical competitive ratios for each error regime. LA-ECT$[\gamma]$ performs very well with perfect predictions (Fig. 5(a)) – with fully trusted predictions, it is nearly 1-competitive against OPT, and all of the learning-augmented algorithms significantly outperform both ZCL and ECT. As the prediction error increases (Figs. 5(b) and 5(c)), the tails of the CDFs degrade accordingly. LA-ECT$[\gamma = 1]$ fully trusts the prediction and has no robustness guarantee – as such, increasing the prediction error induces an unbounded empirical competitive ratio for this case. For other values of $\gamma$, higher error intuitively has a greater impact when the trust parameter $\gamma$ is larger. Notably, LA-ECT$[\gamma = 0.33]$ maintains a worst-case competitive ratio roughly on par with ECT in every error regime while performing better than ZCL and ECT on average across the board.

## 6 CONCLUSION

We study time fairness in the online knapsack problem (OKP), showing impossibility results for existing notions, and proposing a generalizable definition of *conditional time-independent fairness*. We give a deterministic algorithm achieving the Pareto-optimal fairness/efficiency trade-off, and explore the power of randomization and predictions. Evaluating our ECT and LA-ECT algorithms, we observe positive results for competitiveness compared to existing algorithms in exchange for significantly improved fairness guarantees. There are several interesting directions of inquiry that we have not considered which would be good candidates for future work. It would be interesting to apply our notion of time fairness in other online problems such as one-way trading and bin packing, among others (El-Yaniv et al., 2001; Lorenz et al., 2008; Balogh et al., 2017; Johnson et al., 1974; Ma et al., 2021; Balseiro et al., 2023). Another fruitful direction is exploring the notion of *group fairness* (Patel et al., 2020) in OKP, which is practically relevant beyond the settings considered in this paper.

ACKNOWLEDGMENTS

This research is supported by National Science Foundation grants CNS-2102963, CNS-2106299, CPS-2136199, CCF-1908849, NGSDI-2105494, CAREER-2045641, NRT-2021693, and GCR-2020888.

This material is based upon work supported by the U.S. Department of Energy, Office of Science, Office of Advanced Scientific Computing Research, Department of Energy Computational Science Graduate Fellowship under Award Number DE-SC0024386, and Natural Sciences and Engineering Research Council of Canada (NSERC) under grant number DGECR-2018-00059.

DISCLAIMERS

This report was prepared as an account of work sponsored by an agency of the United States Government. Neither the United States Government nor any agency thereof, nor any of their employees, makes any warranty, express or implied, or assumes any legal liability or responsibility for the accuracy, completeness, or usefulness of any information, apparatus, product, or process disclosed, or represents that its use would not infringe privately owned rights. Reference herein to any specific commercial product, process, or service by trade name, trademark, manufacturer, or otherwise does not necessarily constitute or imply its endorsement, recommendation, or favoring by the United States Government or any agency thereof. The views and opinions of authors expressed herein do not necessarily state or reflect those of the United States Government or any agency thereof.

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

SUPPLEMENTARY MATERIAL FOR "TIME FAIRNESS IN ONLINE KNAPSACK PROBLEMS"

Table 1: A summary of key notations.

| Notation | Description |
|---|---|
| $j \in [n]$ | Current position in sequence of items |
| $x_j \in \{0, 1\}$ | Decision for $j$th item. $x_j = 1$ if accepted, $x_j = 0$ if not accepted |
| $U$ | Upper bound on the value density of any item |
| $L$ | Lower bound on the value density of any item |
| $\alpha$ | Fairness parameter, defined in Definition 3.4 |
| $w_j$ | (*Online input*) Item weight revealed to the player when $j$th item arrives |
| $v_j$ | (*Online input*) Item value revealed to the player when $j$th item arrives |

## A  RELATED WORK

We consider the online knapsack problem (OKP), a classical resource allocation problem wherein items with different weights and values arrive sequentially, and we wish to admit them into a capacity-limited knapsack, maximizing the total value subject to the capacity constraint.

Our work contributes to several active lines of research, including a rich literature on OKP first studied in Marchetti-Spaccamela and Vercellis (1995), with important optimal results following in Zhou et al. (2008). In the past several years, research in this area has surged, with many works considering variants of the problem, such as removable items Cygan et al. (2016), item departures Sun et al. (2022), and generalizations to multidimensional settings Yang et al. (2021). Closest to this work, several studies have considered the online knapsack problem with additional information or in a learning-augmented setting, including frequency predictions Im et al. (2021), online learning Zeynali et al. (2021), and advice complexity Böckenhauer et al. (2014).

In the literature on the knapsack problem, fairness has been recently considered in Patel et al. (2020) and Fluschnik et al. (2019). Unlike our work, both deal exclusively with the standard offline setting of knapsack, and consequently do not consider time fairness. (Patel et al., 2020) introduces a notion of *group fairness* for the knapsack problem, while (Fluschnik et al., 2019) introduces three notions of "individually best, diverse, and fair" knapsacks which aggregate the voting preferences of multiple voters. To the best of our knowledge, our work is the first to consider notions of fairness in the online knapsack problem.

In the broader literature on online algorithms and dynamic settings, several studies explore fairness, although most consider notions of fairness that are different from the ones in our work. Online fairness has been explored in resource allocation Manshadi et al. (2021); Sinclair et al. (2020); Bateni et al. (2016); Sinha et al. (2023), fair division Kash et al. (2013), refugee assignment Freund et al. (2023), matchings Deng et al. (2023); Ma et al. (2023), prophet inequalities Correa et al. (2021), organ allocation Bertsimas et al. (2013), and online selection Benomar et al. (2023). Banerjee et al. (2022) shows competitive algorithms for online resource allocation which seek to maximize the Nash social welfare, a metric which quantifies a trade-off between fairness and performance. Deng et al. (2023) also explicitly considers the intersection between predictions and fairness in the online setting. In addition to these problems, there are also several online problems adjacent to OKP in the literature which would be interesting to explore from a fairness perspective, including one-way trading El-Yaniv et al. (2001); Sun et al. (2022), bin packing Balogh et al. (2017); Johnson et al. (1974), and single-leg revenue management Balseiro et al. (2023); Ma et al. (2021).

In the online learning literature, several works consider fairness using regret as a performance metric. In particular, Talebi and Proutiere (2018) studies a stochastic multi-armed bandit setting where tasks must be assigned to servers in a proportionally fair manner. Several other works including Baek and Farias (2021); Patil et al. (2021); Chen et al. (2020) build on these results using different notions of fairness. For a general resource allocation problem, Sinha et al. (2023) presents a fair online resource

allocation policy achieving sublinear regret with respect to an offline optimal allocation. Furthermore, many works in the regret setting explicitly consider fairness from the perspective of an $\alpha$-fair utility function, including Sinha et al. (2023); Si Salem et al. (2022); Wang et al. (2022) , which partially inspires our *parameterized* definition of $\alpha$-CTIF (Def. 3.4).

In the broader ML research community, fairness has seen burgeoning recent interest, particularly from the perspective of bias in trained models. Mirroring the literature above, multiple definitions of fairness in this setting have been proposed and studied, including equality of opportunity Hardt et al. (2016), conflicting notions of fairness Kleinberg et al. (2017), and inherent trade-offs between performance and fairness constraints Bertsimas et al. (2012). Fairness has also been extensively studied in impact studies such as Chouldechova (2017), which demonstrated the disparate impact of recidivism prediction algorithms used in the criminal justice system on different demographics.

## B  PROOFS FOR SECTION 3 (TIME FAIRNESS)

**Observation 3.2.** *The* ZCL *algorithm (Zhou et al., 2008) is not* TIF.

*Proof.* Let $z_j \in [0, 1]$ be the knapsack's utilization when the $j$th item arrives. When the knapsack is empty, $z_j = 0$, and when the knapsack is close to full, $z_j = 1 - \epsilon$ for some small $\epsilon > 0$. Pick any instance with sufficiently many items, and pick $z_A < z_B$ such that at least one admitted item, say the $k$th one, satisfies $\Phi(z_A) \le v_k < \Phi(z_B)$. Note that this implies that the $k$th item arrived between the utilization being at $z_A$ and at $z_B$. Now, modify this same instance by adding a copy of the $k$th item after the utilization has reached $z_B$. Note that this item now has probability zero of being admitted. This implies that two items with the same value-to-weight ratio have different probabilities of being admitted into the knapsack, contradicting the definition of TIF. □

**Theorem 3.3.** *There is no nontrivial algorithm for* OKP *that guarantees* TIF *without additional information about the input. Further, even if the input length $n$ or perfect frequency predictions as defined in Im et al. (2021) are known in advance, no nontrivial algorithm can guarantee* TIF.

*Proof.* We prove the three parts one by one.

1. WLOG assume that all items have the same value density $x = L = U$. Assume for the sake of a contradiction that there is some algorithm ALG which guarantees TIF, and suppose we have a instance $\mathcal{I}$ with $n$ items (which we will set later).

   Since we assume that ALG guarantees TIF, consider $p(x)$, the probability of admitting an item with value density $x$. Let $p(x) = \mathrm{p}$. Note that $\mathrm{p} > 0$, and also that it cannot depend on the input length $n$. Note that we can have have all items of equal weight $w$, so that for $n \gg 3B/w\mathrm{p}$, the knapsack is full with high probability. But now consider modifying the input sequence from $\mathcal{I}$ to $\mathcal{I}'$, by appending a copy of itself, i.e., increasing the input with another $n$ items of exactly the same type. The probability of admitting these new items must be (eventually) zero, even though they have the same value-to-weight ratio as the first half of the input (and, indeed, the same weights). Therefore, the algorithm violates the TIF constraint on the instance $\mathcal{I}'$, which is a contradiction.

2. Again WLOG assume that all items have the same value density $x = L = U$, so that $s_x = \sum_{i=1}^{n} w_i$. Assume for the sake of a contradiction that there is some competitive algorithm $\mathsf{ALG}^{\mathrm{FP}}$ which uses frequency predictions to guarantee TIF.

   Since we assume that $\mathsf{ALG}^{\mathrm{FP}}$ guarantees TIF, consider $p(x)$, the probability of admitting any item. Let $p(x) = \mathrm{p}$. Again, note that $\mathrm{p} > 0$, and also that it cannot depend on the input length $n$, but can depend on $s_x$ this time.

   Consider two instances $\mathcal{I}$ and $\mathcal{I}'$ as follows: $\mathcal{I}$ consists exclusively of "small" items of (constant) weight $w_\delta \ll B$ each, whereas $\mathcal{I}'$ consists of a small, constant number of such small items followed by a single "large" item of weight $B - \delta/2$. Note that by taking enough items in $\mathcal{I}$, we can ensure the two instances have the same total weight $s(x)$. Therefore, $p(x)$ must be the same for these two instances, by assumption. Of course, the items all have value $x$ by our original assumption.

Note that the optimal packing in both instances would nearly fill up the knapsack. However, $\mathcal{I}'$ has the property that any competitive algorithm must reject all the initial smaller items, as admitting any of them would imply that the large item can no longer be admitted. By making $w_\delta$ arbitrarily small, we can make the algorithm arbitrarily non-competitive.

The instance $\mathcal{I}$ guarantees that $p(x)$ is sufficiently large (i.e., bounded below by some constant), and so with high probability, at least one item in $\mathcal{I}'$ is admitted within the first constant number of items. Therefore, with high probability, $\mathsf{ALG}^{\mathrm{FP}}$ does not admit the large-valued item in $\mathcal{I}'$, and so it cannot be competitive.

3. Again WLOG assume that all items have the same value density $x = L = U$. Assume for the sake of a contradiction that there is some competitive algorithm $\mathsf{ALG}^{\mathrm{N}}$ which uses knowledge of the input length $n$ to guarantee TIF.

   We will consider only input sequences of length $n$ (assumed to be sufficiently large), consisting only of items with value density $x$. Again, since we assume that $\mathsf{ALG}^{\mathrm{FP}}$ guarantees TIF, consider $p(x)$, the probability of admitting any item. Let $p(x) = \mathrm{p}$. Again, note that $\mathrm{p} > 0$, and also that it must be the same for all input sequences of length $n$.

   Consider such an instance $\mathcal{I}$, consisting of identical items of (constant) weight $w_c$ each. Suppose the total weight of the items is very close to the knapsack capacity $B$. Since the expected number of items admitted is $n\mathrm{p}$, the total value admitted is $x \cdot n\mathrm{p}$ on expectation. The optimal solution admits a total value of $nx$ (since the total weight is close to $B$), and therefore, the competitive ratio is roughly $1/\mathrm{p}$. Since we assumed the algorithm was competitive, it follows that $\mathrm{p}$ must be bounded below by a constant.

   Now consider a different instance $\mathcal{I}'$, consisting of $3\log(n)/\mathrm{p}^2$ items of weight $w_\delta \ll B$, followed by $n - 3\log(n)/\mathrm{p}^2$ "large" items of weight $B - w_\delta/2$. Note that these are well-defined, as $\mathrm{p}$ is bounded below by a constant, and $n$ is sufficiently large. The instance $\mathcal{I}'$ again has the property that any competitive algorithm must reject all the initial smaller items, as admitting any of them would imply that none of the large items can be admitted.

   However, by the coupon collector problem, with high probability $(1 - \mathrm{poly}(1/n))$, at least one of the $3\log(n)/\mathrm{p}^2$ small items is admitted, which contradicts the competitiveness of $ALG^{\mathrm{N}}$. As before, by making $w_\delta$ arbitrarily small, we can make the algorithm arbitrarily non-competitive. $\qquad\square$

**Observation 3.5.** *The* $\mathsf{ZCL}$ *algorithm (Zhou et al., 2008) is not* 1*-CTIF.*

*Proof.* This follows immediately from the fact that the threshold value in ZCL changes each time an item is accepted, which corresponds to the utilization changing. Consider two items with the same value density (close to $L$), where one of the items arrives first in the sequence, and the other arrives when the knapsack is roughly half-full, and assume that there is enough space in the knapsack to accommodate both items when they arrive. The earlier item will be admitted with certainty, whereas the later item will with high probability be rejected. So despite having the same value, the items will have a different admission probability purely based on their position in the sequence, violating 1-CTIF. $\qquad\square$

## C  PROOFS FOR SECTION 4 (ONLINE FAIR ALGORITHMS)

**Proposition 4.1.** *Any constant threshold-based algorithm for* $\mathsf{OKP}$ *is* 1*-CTIF. Furthermore, any constant threshold-based* deterministic *algorithm for* $\mathsf{OKP}$ *cannot be better than* $(U/L)$*-competitive.*

*Proof.* Consider an arbitrary threshold-based algorithm ALG with constant threshold value $\phi$. For any instance $\mathcal{I}$, and any item, say the $j$th one, in this instance, note that the probability of admitting the item depends entirely on the threshold $\phi$, and nothing else, as long there is enough space in the knapsack to admit it. So for any value density $x \in [L, U]$, the admission probability $p(x)$ is just the indicator variable capturing whether there is space for the item or not.

For the second part, given a deterministic ALG with a fixed constant threshold $\phi \in [L, U]$, there are two cases. If $\phi > L$, the instance $\mathcal{I}$ consisting entirely of $L$-valued items induces an unbounded

competitive ratio, as no items are admitted by ALG. If $\phi = L$, consider the instance $\mathcal{I}'$ consisting of $m$ equal-weight items with value $L$ followed by $m$ items with value $U$, and take $m$ large enough that the knapsack can become full with only $L$-valued items. ALG here admits only $L$-valued items, whereas the optimal solution only admits $U$-valued items, and so ALG cannot do better than the worst-case competitive ratio of $\frac{U}{L}$ for OKP. $\qquad\square$

**Proposition 4.2.** *The* ZCL *algorithm is* $\frac{1}{\ln(U/L)+1}$-CTIF.

*Proof.* Consider the interval $\left[0, \frac{1}{\ln(\frac{U}{L})+1}\right]$, viewed as an utilization interval. An examination of the ZCL algorithm reveals that the value of the threshold is below $L$ on this subinterval. But since we have a guarantee that the value-to-weight ratio is at least $L$, while the utilization is within this interval, the ZCL algorithm is exactly equivalent to the algorithm using the constant threshold $L$. By Proposition 4.1, therefore, the algorithm is 1-CTIF within this interval, and therefore is $\frac{1}{\ln(\frac{U}{L})+1}$-CTIF. $\qquad\square$

**Theorem 4.3.** *For* $\alpha \in [1/\ln(U/L)+1, 1]$*, the baseline algorithm is* $\frac{U[\ln(U/L)+1]}{L\alpha[\ln(U/L)+1]+(U-L)(1-\ell)}$-*competitive and* $\alpha$-CTIF *for* OKP.

*Proof.* To prove the competitive ratio of the parameterized baseline algorithm (call it BASE$[\alpha]$), consider the following:

Fix an arbitrary instance $\mathcal{I} \in \Omega$. When the algorithm terminates, suppose the utilization of the knapsack is $z_T$. Assume we obtain a value of BASE$[\alpha](\mathcal{I})$. Let $\mathcal{P}$ and $\mathcal{P}^\star$ respectively be the sets of items picked by BASE$[\alpha]$ and the optimal solution.

Denote the weight and the value of the common items (i.e., the items picked by both BASE and OPT) by $W = w(\mathcal{P} \cap \mathcal{P}^\star)$ and $V = v(\mathcal{P} \cap \mathcal{P}^\star)$. For each item $j$ which is *not accepted* by BASE$[\alpha]$, we know that its value density is $< \Phi^\alpha(z_j) \le \Phi^\alpha(z_T)$ since $\Phi^\alpha$ is a non-decreasing function of $z$. Thus, using $B = 1$, we get

$$\mathsf{OPT}(\mathcal{I}) \le V + \Phi^\alpha(z_T)(1 - W).$$

Since BASE$[\alpha](\mathcal{I}) = V + v(\mathcal{P} \setminus \mathcal{P}^\star)$, the inequality above implies that

$$\frac{\mathsf{OPT}(\mathcal{I})}{\mathsf{BASE}[\alpha](\mathcal{I})} \le \frac{V + \Phi^\alpha(z_T)(1 - W)}{V + v(\mathcal{P} \setminus \mathcal{P}^\star)}.$$

Note that, by definition of the algorithm, each item $j$ picked in $\mathcal{P}$ must have value density of at least $\Phi^\alpha(z_j)$, where $z_j$ is the knapsack utilization when that item arrives. Thus, we have:

$$V \ge \sum_{j \in \mathcal{P} \cap \mathcal{P}^\star} \Phi^\alpha(z_j)\, w_j =: V_1,$$

$$v(\mathcal{P} \setminus \mathcal{P}^\star) \ge \sum_{j \in \mathcal{P} \setminus \mathcal{P}^\star} \Phi^\alpha(z_j)\, w_j =: V_2.$$

Since $\mathsf{OPT}(\mathcal{I}) \ge \mathsf{BASE}[\alpha](\mathcal{I})$, we have:

$$\frac{\mathsf{OPT}(\mathcal{I})}{\mathsf{BASE}[\alpha](\mathcal{I})} \le \frac{V + \Phi^\alpha(z_T)(1 - W)}{V + v(\mathcal{P} \setminus \mathcal{P}^\star)} \le \frac{V_1 + \Phi^\alpha(z_T)(1 - W)}{V_1 + v(\mathcal{P} \setminus \mathcal{P}^\star)} \le \frac{V_1 + \Phi^\alpha(z_T)(1 - W)}{V_1 + V_2},$$

where the second inequality follows because $\Phi^\alpha(z_T)(1 - W) \ge v(\mathcal{P} \setminus \mathcal{P}^\star)$ and $V_1 \le V$.

Note that $V_1 \le \Phi^\alpha(z_T)w(\mathcal{P} \cap \mathcal{P}^\star) = \Phi^\alpha(z_T)W$, and by plugging in the actual values of $V_1$ and $V_2$, we get:

$$\frac{\mathsf{OPT}(\mathcal{I})}{\mathsf{BASE}[\alpha](\mathcal{I})} \le \frac{\Phi^\alpha(z_T)}{\sum_{j \in \mathcal{P}} \Phi^\alpha(z_j)\, w_j}.$$

Based on the assumption that individual item weights are much smaller than 1, we can substitute for $w_j$ with $\delta z_j = z_{j+1} - z_j$ for all $j$. This substitution gives an approximate value of the summation via integration.

$$
\sum_{j \in \mathcal{P}} \Phi^\alpha(z_j)\, w_j \approx \int_0^{z_T} \Phi^\alpha(z) dz
$$

$$
= \int_0^\alpha L\, dz + \int_\alpha^{z_T} \left(\frac{Ue}{L}\right)^{\frac{z-\ell}{1-\ell}} \left(\frac{L}{e}\right) dz
$$

$$
= L\alpha + \left(\frac{L}{e}\right)(1-\ell)\left[\frac{\left(\frac{Ue}{L}\right)^{\frac{z-\ell}{1-\ell}}}{\ln(Ue/L)}\right]_\alpha^{z_T}
$$

$$
= L\alpha + \frac{\Phi^\alpha(z_T)}{\ln(Ue/L)} - \frac{\Phi^\alpha(\alpha)}{\ln(Ue/L)} - \frac{\ell\Phi^\alpha(z_T)}{\ln(Ue/L)} + \frac{\ell\Phi^\alpha(\alpha)}{\ln(Ue/L)}
$$

$$
= L\left(\alpha - \frac{1}{\ln(Ue/L)} + \frac{\ell}{\ln(Ue/L)}\right) + \frac{\Phi^\alpha(z_T)}{\ln(Ue/L)} - \frac{\ell\Phi^\alpha(z_T)}{\ln(Ue/L)}
$$

$$
= L\left(\alpha - \frac{1-\ell}{\ln(U/L)+1}\right) + \frac{(1-\ell)\Phi^\alpha(z_T)}{\ln(U/L)+1},
$$

where the fourth equality has used the fact that $\Phi^\alpha(z_j) = (L/e)(Ue/L)^{\frac{z-\ell}{1-\ell}}$, and the fifth equality has used $\Phi^\alpha(\alpha) = L$. Substituting back in, we get:

$$
\frac{\mathsf{OPT}(\mathcal{I})}{\mathsf{BASE}[\alpha](\mathcal{I})} \leq \frac{\Phi^\alpha(z_T)}{L\left(\alpha - \frac{1-\ell}{\ln(U/L)+1}\right) + \frac{(1-\ell)\Phi^\alpha(z_T)}{\ln(U/L)+1}} \leq \frac{U[\ln(U/L)+1]}{L\alpha[\ln(U/L)+1] - L(1-\ell) + U(1-\ell)}.
$$

Thus, the baseline algorithm is $\frac{U[\ln(U/L)+1]}{L\alpha[\ln(U/L)+1] - L(1-\ell) + U(1-\ell)}$-competitive.

The fairness constraint of $\alpha$-CTIF is immediate, because the threshold $\Phi^\alpha(z) \leq L$ in the interval $[0, \alpha]$, and so it can be replaced by the constant threshold $L$ in that interval. Applying Proposition 4.1 yields the result. $\qquad\square$

**Theorem 4.5.** *No $\alpha$-CTIF deterministic online algorithm for* OKP *can achieve a competitive ratio smaller than* $\frac{W\left(\frac{U(1-\alpha)}{L\alpha}\right)}{1-\alpha}$, *where $W(\cdot)$ is the Lambert W function.*

*Proof.* For any $\alpha$-CTIF deterministic online algorithm ALG, there must exist some utilization region $[a, b]$ with $b - a = \alpha$. Any item that arrives in this region is treated fairly, i.e., by definition of CTIF there exists a function $p(x) : [L, U] \to \{0, 1\}$ which characterizes the fair decisions of ALG. We define $v = \min\{x \in [L, U] : p(x) = 1\}$ (i.e., $v$ is the lowest value density that ALG is willing to accept during the fair region).

We first state a lemma (proven afterwards), which asserts that the feasible competitive ratio for any $\alpha$-CTIF deterministic online algorithm with $v > L$ is *strictly worse* than the feasible competitive ratio when $v = L$.

**Lemma C.1.** *For any $\alpha$-CTIF deterministic online algorithm* ALG *for* OKP*, if the minimum value density $v$ that* ALG *accepts during the fair region of the utilization (of length $\alpha$) is $> L$, then it must have a competitive ratio $\beta' \geq W\left(\frac{U(1-\alpha)}{L\alpha}e^{\frac{1}{\alpha}}\right)/(1-\alpha) - \frac{1}{\alpha}$, where $W(\cdot)$ is the Lambert W function.*

By Lemma C.1, it suffices to consider the algorithms that set $v = L$.

Given ALG, let $g(x) : [L, U] \to [0, 1]$ denote the *acceptance function* of ALG, where $g(x)$ is the final knapsack utilization under the instance $\mathcal{I}_x$. Note that for small $\delta$, processing $\mathcal{I}_{x+\delta}$ is equivalent to first processing $\mathcal{I}_x$, and then processing $m$ identical items, each with weight $\frac{1}{m}$ and value density $x + \delta$. Since this function is unidirectional (item acceptances are irrevocable) and deterministic, we must have $g(x + \delta) \geq g(x)$, i.e. $g(x)$ is non-decreasing in $[L, U]$. Once a batch of items with maximum value density $U$ arrives, the rest of the capacity should be used, i.e., $g(U) = 1$.

For any algorithm with $v = L$, it will admit *all* items greedily for an $\alpha$ fraction of the knapsack. Therefore, under the instance $\mathcal{I}_x$, the online algorithm with acceptance function $g$ obtains a value of $\mathsf{ALG}(\mathcal{I}_x) = \alpha L + g(r)r + \int_r^x u dg(u)$, where $u dg(u)$ is the value obtained by accepting items with value density $u$ and total weight $dg(u)$, and $r$ is defined as the lowest value density that $\mathsf{ALG}$ is willing to accept during the unfair region, i.e., $r = \inf_{x \in (L,U): g(x) \geq \alpha} x$.

For any $\beta'$-competitive algorithm, we must have $r \leq \alpha \beta' L$ since otherwise the worst-case ratio is larger than $\beta'$ under an instance $\mathcal{I}_x$ with $x = \alpha \beta' L + \epsilon$, ($\epsilon > 0$). To derive a lower bound of the competitive ratio, observe that it suffices WLOG to focus on algorithms with $r = \alpha \beta' L$. This is because if a $\beta'$-competitive algorithm sets $r < \alpha \beta' L$, then an alternative algorithm can postpone the item acceptance to $r = \alpha \beta' L$ and maintain $\beta'$-competitiveness.

Under the instance $\mathcal{I}_x$, the offline optimal solution obtains a total value of $\mathsf{OPT}(\mathcal{I}_x) = x$. Therefore, any $\beta'$-competitive online algorithm must satisfy:

$$\mathsf{ALG}(\mathcal{I}_x) = g(L)L + \int_L^x u dg(u) = \alpha L + \int_r^x u dg(u) \geq \frac{x}{\beta'}, \quad \forall x \in [L, U].$$

By integral by parts and Grönwall's Inequality (Theorem 1, p. 356, in Mitrinovic et al. (1991)), a necessary condition for the competitive constraint above to hold is the following:

$$g(x) \geq \frac{1}{\beta'} - \frac{\alpha L - g(r)r}{x} + \frac{1}{x} \int_r^x g(u) du \tag{2}$$

$$\geq \frac{1}{\beta'} - \frac{\alpha L}{r} + g(r) + \frac{1}{\beta'} \ln \frac{x}{r} = g(\alpha \beta' L) + \frac{1}{\beta'} \ln \frac{x}{\alpha \beta' L}, \quad \forall x \in [L, U]. \tag{3}$$

By combining $g(U) = 1$ with equation (3), it follows that any deterministic $\alpha$-CTIF and $\beta'$-competitive algorithm must satisfy $1 = g(U) \geq g(\alpha \beta' L) + \frac{1}{\beta'} \ln \frac{x}{\alpha \beta' L} \geq \alpha + \frac{1}{\beta'} \ln \frac{x}{\alpha \beta' L}$. The minimal value for $\beta'$ can be achieved when both inequalities are tight, and is the solution to $\ln(\frac{U}{\alpha \beta' L})/\beta' = 1 - \alpha$. Thus, $\frac{W\left(\frac{U - U\alpha}{L\alpha}\right)}{1 - \alpha}$ is a lower bound of the competitive ratio, where $W(\cdot)$ denotes the Lambert $W$ function. $\square$

**Proof of Lemma C.1** We use the same definition of the acceptance function $g(x)$ as that in Theorem 4.5. Based on the choice of $v$ by $\mathsf{ALG}$, we consider the following two cases.

**Case I: when $v \geq \frac{U}{1 + \alpha \beta'}$.** Under the instance $\mathcal{I}_x$ with $x \in [L, v)$, the offline optimum is $\mathsf{OPT}(\mathcal{I}_x) = x$ and $\mathsf{ALG}$ can achieve $\mathsf{ALG}(\mathcal{I}_x) = Lg(L) + \int_L^x u dg(u)$. Thus, any $\beta'$-competitive algorithm must satisfy:

$$\mathsf{ALG}(\mathcal{I}_x) = Lg(L) + \int_L^x u dg(u) \geq \frac{x}{\beta'}, \quad x \in [L, v).$$

By integral by parts and Grönwall's Inequality (Theorem 1, p. 356, in Mitrinovic et al. (1991)), a necessary condition for the inequality above to hold is:

$$g(x) \geq \frac{1}{\beta'} + \frac{1}{x} \int_L^x g(u) du \geq \frac{1}{\beta'} + \frac{1}{\beta'} \ln \frac{x}{L}, \forall x \in [L, v).$$

Under the instance $\mathcal{I}_v$, to maintain $\alpha$-CTIF , we must have $g(v) \geq \lim_{x \to v}[\frac{1}{\beta'} + \frac{1}{\beta'} \ln \frac{x}{L}] + \alpha = \frac{1}{\beta'} + \frac{1}{\beta'} \ln \frac{v}{L} + \alpha$. Thus, we have $1 \geq g(v) \geq \frac{1}{\beta'} + \frac{1}{\beta'} \ln \frac{v}{L} + \alpha$, which gives:

$$\beta' \geq \frac{1 + \ln \frac{v}{L}}{1 - \alpha}. \tag{4}$$

This lower bound is achieved when $g(x) = \frac{1}{\beta'} + \frac{1}{\beta'} \ln \frac{x}{L}, x \in [L, v)$ and $g(v) = \frac{1}{\beta'} + \frac{1}{\beta'} \ln \frac{x}{L} + v$. In addition, the total value of accepted items is $\mathsf{ALG}(\mathcal{I}_v) = \frac{v}{\beta'} + \alpha v$.

Under the instance $\mathcal{I}_x$ with $x \in (v, U]$, we observe that the worst-case ratio is:

$$\frac{\mathsf{OPT}(\mathcal{I}_x)}{\mathsf{ALG}(\mathcal{I}_x)} \leq \frac{U}{\mathsf{ALG}(\mathcal{I}_v)} = \beta' \frac{U}{v(1 + \alpha \beta')} \leq \beta'.$$

Thus, the lower bound of the competitive ratio is dominated by equation (4), and $\beta' \geq \frac{1 + \ln \frac{v}{L}}{1 - \alpha} \geq \frac{1 + \ln \frac{U}{L(1 + \alpha \beta')}}{1 - \alpha}$.

**Case II: when $L < v < \frac{U}{1+\alpha\beta'}$.** In this case, we have the same results under instances $\mathcal{I}_x, x \in [L, v]$. In particular, $g(v) = \frac{1}{\beta'} + \frac{1}{\beta'} \ln \frac{x}{L} + v$ and $\mathsf{ALG}(\mathcal{I}_v) = \frac{v}{\beta'} + \alpha v$.

Under the instance $\mathcal{I}_x$ with $x \in (v, U]$, the online algorithm can achieve

$$\mathsf{ALG}(\mathcal{I}_x) = \mathsf{ALG}(\mathcal{I}_v) + \int_r^x u dg(u), x \in (v, U], \tag{5}$$

where $r$ is the lowest value density that ALG admits outside of the fair region. Using the same argument as that in the proof of Theorem 4.5, WLOG we can consider $r = \beta' \cdot \mathsf{ALG}(\mathcal{I}_v) = v(1 + \alpha\beta'), (r < U)$. By integral by parts and Grönwall's Inequality, a necessary condition for equation (5) is:

$$
\begin{aligned}
g(x) &\geq \frac{1}{\beta'} - \frac{\mathsf{ALG}(\mathcal{I}_v) - g(r)r}{x} + \frac{1}{x} \int_r^x g(u) du \\
&\geq \frac{1}{\beta'} - \frac{\mathsf{ALG}(\mathcal{I}_v) - g(r)r}{r} + \frac{1}{\beta'} \ln \frac{x}{r} \\
&= g(r) + \frac{1}{\beta'} \ln \frac{x}{r}.
\end{aligned}
$$

Combining with $g(U) = 1$ and $g(r) \geq g(v)$, we have:

$$1 = g(U) \geq g(r) + \frac{1}{\beta'} \ln \frac{U}{r} \geq \alpha + \frac{1}{\beta'} + \frac{1}{\beta'} \ln \frac{v}{L} + \frac{1}{\beta'} \ln \frac{U}{v(1 + \alpha\beta')},$$

and thus, the competitive ratio must satisfy:

$$\beta' \geq \frac{1 + \ln \frac{U}{L(1+\alpha\beta')}}{1 - \alpha}. \tag{6}$$

Recall that under the instance $\mathcal{I}_x$ with $x \in [L, v)$, the worst-case ratio is:

$$\frac{\mathsf{OPT}(\mathcal{I}_x)}{\mathsf{ALG}(\mathcal{I}_x)} \leq \frac{1 + \ln \frac{v}{L}}{1 - \alpha} < \frac{1 + \ln \frac{U}{L(1+\alpha\beta')}}{1 - \alpha}.$$

Therefore, the lower bound is dominated by equation (6).

Summarizing above two cases, for any $\alpha$-CTIF deterministic online algorithm, if the minimum value density $v$ that it is willing to accept during the fair region is $> L$, then its competitive ratio must satisfy $\beta' \geq \frac{1+\ln \frac{U}{L(1+\alpha\beta')}}{1-\alpha}$, and the lower bound of the competitive ratio is $\frac{W\left(\frac{(1-\alpha)U}{L\alpha} e^{1/\alpha}\right)}{1-\alpha}$. It is also easy to verify that $\frac{W\left(\frac{(1-\alpha)U}{L\alpha} e^{1/\alpha}\right)}{1-\alpha} \geq \frac{W\left(\frac{U-U\alpha}{L\alpha}\right)}{1-\alpha}, \forall \alpha \in [\frac{1}{\ln(U/L)+1}, 1]$. Thus, for any $\alpha$-CTIF algorithm, we focus on the algorithms where $v = L$ in order to minimize the competitive ratio. $\square$

**Theorem 4.6.** *For any $\alpha \in [1/\ln(U/L)+1, 1]$, $\mathsf{ECT}[\alpha]$ is $\beta$-competitive and $\alpha$-CTIF.*

*Proof.* Fix an arbitrary instance $\mathcal{I} \in \Omega$. When $\mathsf{ECT}[\alpha]$ terminates, suppose the utilization of the knapsack is $z_T$, and assume we obtain a value of $\mathsf{ECT}[\alpha](\mathcal{I})$. Let $\mathcal{P}$ and $\mathcal{P}^\star$ respectively be the sets of items picked by $\mathsf{ECT}[\alpha]$ and the optimal solution.

Denote the weight and the value of the common items (i.e., the items picked by both ECT and OPT) by $W = w(\mathcal{P} \cap \mathcal{P}^\star)$ and $V = v(\mathcal{P} \cap \mathcal{P}^\star)$. For each item $j$ which is *not accepted* by $\mathsf{ECT}[\alpha]$, we know that its value density is $< \Psi^\alpha(z_j) \leq \Psi^\alpha(z_T)$ since $\Psi^\alpha$ is a non-decreasing function of $z$. Thus,

$$\mathsf{OPT}(\mathcal{I}) \leq V + \Psi^\alpha(z_T)(1 - W).$$

Since $\mathsf{ECT}[\alpha](\mathcal{I}) = V + v(\mathcal{P} \setminus \mathcal{P}^\star)$, the above inequality implies that

$$\frac{\mathsf{OPT}(\mathcal{I})}{\mathsf{ECT}[\alpha](\mathcal{I})} \leq \frac{V + \Psi^\alpha(z_T)(1 - W)}{V + v(\mathcal{P} \setminus \mathcal{P}^\star)}.$$

Note that, by definition of the algorithm, each item $j$ picked in $\mathcal{P}$ must have value density at least $\Psi^\alpha(z_j)$, where $z_j$ is the knapsack utilization when that item arrives. Thus, we have:

$$V \geq \sum_{j \in \mathcal{P} \cap \mathcal{P}^\star} \Psi^\alpha(z_j)\, w_j =: V_1,$$

$$v(\mathcal{P} \setminus \mathcal{P}^\star) \geq \sum_{j \in \mathcal{P} \setminus \mathcal{P}^\star} \Psi^\alpha(z_j)\, w_j =: V_2.$$

Since $\mathsf{OPT}(\mathcal{I}) \geq \mathsf{ECT}[\alpha](\mathcal{I})$, we have:

$$\frac{\mathsf{OPT}(\mathcal{I})}{\mathsf{ECT}[\alpha](\mathcal{I})} \leq \frac{V + \Psi^\alpha(z_T)(1-W)}{V + v(\mathcal{P} \setminus \mathcal{P}^\star)} \leq \frac{V_1 + \Psi^\alpha(z_T)(1-W)}{V_1 + v(\mathcal{P} \setminus \mathcal{P}^\star)} \leq \frac{V_1 + \Psi^\alpha(z_T)(1-W)}{V_1 + V_2},$$

where the second inequality follows because $\Psi^\alpha(z_T)(1-W) \geq v(\mathcal{P} \setminus \mathcal{P}^\star)$ and $V_1 \leq V$.

Note that $V_1 \leq \Psi^\alpha(z_T) w(\mathcal{P} \cap \mathcal{P}^\star) = \Psi^\alpha(z_T) W$, and by plugging in for the actual values of $V_1$ and $V_2$ we get:

$$\frac{\mathsf{OPT}(\mathcal{I})}{\mathsf{ECT}[\alpha](\mathcal{I})} \leq \frac{\Psi^\alpha(z_T)}{\sum_{j \in \mathcal{P}} \Psi^\alpha(z_j)\, w_j}.$$

Based on the assumption that individual item weights are much smaller than 1, we can substitute for $w_j$ with $\delta z_j = z_{j+1} - z_j$ for all $j$. This substitution gives an approximate value of the summation via integration:

$$\sum_{j \in \mathcal{P}} \Psi^\alpha(z_j)\, w_j \approx \int_0^{z_T} \Psi^\alpha(z) dz$$

Now there are two cases to analyze – the case where $z_T = \alpha$, and the case where $z_T > \alpha$. Note that $z_T < \alpha$ is impossible, as this means $\mathsf{ECT}[\alpha]$ rejected some item that it had capacity for even when the threshold was at $L$, which is a contradiction. We explore each of the two cases in turn below.

**Case I.** If $z_T = \alpha$, then $\sum_{j \in \mathcal{P}} \Psi^\alpha(z_j)\, w_j \geq L\alpha$.
This follows because $\mathsf{ECT}[\alpha]$ is effectively greedy for at least $\alpha$ utilization of the knapsack, and so the admitted items during this portion must have value at least $L\alpha$. Substituting into the original equation gives us the following:

$$\frac{\mathsf{OPT}(\mathcal{I})}{\mathsf{ECT}[\alpha](\mathcal{I})} \leq \frac{\Psi^\alpha(z_T)}{L\alpha} \leq \beta.$$

**Case II.** If $z_T > \alpha$, then $\sum_{j \in \mathcal{P}} \Psi^\alpha(z_j)\, w_j \approx \int_0^\alpha L dz + \int_\alpha^{z_T} \Psi^\alpha(z) dz$.
Solving for the integration, we obtain the following:

$$\sum_{j \in \mathcal{P}} \Psi^\alpha(z_j)\, w_j \approx \int_0^\alpha L dz + \int_\alpha^{z_T} \Psi^\alpha(z) dz$$

$$= L\alpha + \int_\alpha^{z_T} U e^{\beta(z-1)} dz = L\alpha + \left[ \frac{U e^{\beta(z-1)}}{\beta} \right]_\alpha^{z_T}$$

$$= L\alpha - \frac{U e^{\beta(\alpha-1)}}{\beta} + \frac{U e^{\beta(z_T-1)}}{\beta} = L\alpha - \frac{\Psi^\alpha(\alpha)}{\beta} + \frac{\Psi^\alpha(z_T)}{\beta} = \frac{\Psi^\alpha(z_T)}{\beta}.$$

Substituting into the original equation, we can bound the competitive ratio:

$$\frac{\mathsf{OPT}(\mathcal{I})}{\mathsf{ECT}[\alpha](\mathcal{I})} \leq \frac{\Psi^\alpha(z_T)}{\frac{1}{\beta} \Psi^\alpha(z_T)} = \beta,$$

and the result follows.

Furthermore, the value of $\beta$ which solves the equation $x = \frac{U e^{x(\alpha-1)}}{L\alpha}$ can be shown as $\frac{W\left( \frac{U - U\alpha}{L\alpha} \right)}{1-\alpha}$, which matches the lower bound from Theorem 4.5. $\qquad \square$

**Theorem 4.11.** *For any $\gamma \in (0,1]$, and any $\mathcal{I} \in \Omega$, $\mathsf{LA\text{-}ECT}[\gamma]$ is $\frac{2}{\gamma}$-consistent.*

*Proof.* For consistency, assume that the black-box predictor $\rho_{\mathcal{I}}^{-1}(y)$ is accurate (i.e. $\hat{d}_\gamma = d_\gamma^\star$). Let $\epsilon$ denote the upper bound on any individual item's weight (previously assumed to be small).

In Lemma C.2, we describe $\mathsf{ORACLE}_\gamma^\star$, a competitive *semi-online algorithm* (Seiden et al., 2000; Tan and Wu, 2007; Kumar et al., 2019; Dwibedy and Mohanty, 2022) which is restricted to use a knapsack of size $\gamma$. Plainly, it is an algorithm that has full knowledge of the items in the instance, but must process items sequentially using a threshold it has to set in advance. Items still arrive in an online manner, decisions are immediate and irrevocable, and the order of arrival is unknown.

**Lemma C.2.** *There is a deterministic semi-online algorithm,* $\mathsf{ORACLE}_\gamma^\star$, *which is* 1-CTIF, *fills a knapsack of size* $\gamma \in [0, 1]$, *and has an approximation factor of* $2/(\gamma - \epsilon)$. *Moreover, no deterministic semi-online algorithm with an approximation factor less than* $2 - L/U$ *is* 1-CTIF.

**Proof of Lemma C.2**
**Upper bound:** Note that $\mathsf{ORACLE}_\gamma^\star$ can compute $d_\gamma^\star$ before any items arrive. Suppose $\mathsf{ORACLE}_\gamma^\star$ sets its threshold at $d_\gamma^\star$, and therefore admits any items with value density at or above $d_\gamma^\star$.

Recall the definition of the critical threshold $d_\gamma^\star$ from Definition 4.9. For an arbitrary instance $\mathcal{I}$, let $V$ denote the value obtained by $\mathsf{OPT}(\mathcal{I})$, and $d_\gamma^\star$ gives the maximum value density such that the total value of items with value density $\geq d_\gamma^\star$ in $\mathsf{OPT}(\mathcal{I})$ is at least $\gamma V/2$.

Based on the definition of $d_\gamma^\star$, we know that either the total weight of items with value density $\geq d_\gamma^\star$ in $\mathsf{OPT}(\mathcal{I})$'s knapsack is strictly less than $\gamma$, or $\gamma d_\gamma^\star \geq \gamma V/2$. To verify this, we start by sorting $\mathsf{OPT}(\mathcal{I})$'s packed items in non-increasing order of value density.

Suppose a greedy approximation algorithm $\mathsf{APX}_\gamma$ iterates over this list in sorted order, packing items into a knapsack of size $\gamma$ until it is full. Note that $\mathsf{APX}_\gamma$ packs $(\gamma - \epsilon)$ of the highest value density items from $\mathcal{I}$ into its knapsack.

By definition, $\mathsf{APX}_\gamma \geq (\gamma - \epsilon) \cdot V$. In the worst-case, where all items in $\mathsf{OPT}(\mathcal{I})$'s knapsack are the same value density, we have that $\mathsf{APX}_\gamma = (\gamma - \epsilon) \cdot V$.

Denote the value density of the last item packed by $\mathsf{APX}_\gamma$ as $\bar{d}$. For the sake of contradiction, assume that $d_\gamma^\star < \bar{d}$. Since $\mathsf{APX}_\gamma$ fills a $(\gamma - \epsilon)$ fraction of its knapsack with items of value density $\geq \bar{d}$ and obtains a value of at least $(\gamma - \epsilon)V > \frac{\gamma V}{2}$, this causes a contradiction: $d_\gamma^\star$ should be the *largest* value density such that the total value of items with value density $\geq d_\gamma^\star$ in $\mathsf{OPT}(\mathcal{I})$ is at least $\gamma V/2$, but the assumption $d_\gamma^\star < \bar{d}$ implies that the total value of items with value density $\geq \bar{d}$ is also at least $\gamma V/2$.

This further implies either of the following: (I) The true value of $d_\gamma^\star$ is $d_\gamma^\star > \bar{d}$ if $\mathsf{APX}_\gamma$'s knapsack contains enough items with value density $> \bar{d}$ and total weight $< (\gamma - \epsilon)$ such that their total value is at least $\gamma V/2$. (II) $d_\gamma^\star = \bar{d}$ if there are enough items of value density $\bar{d}$ in $\mathcal{I}$ such that $\gamma \bar{d} \geq \gamma V/2$.

Given this information, there are two possible outcomes for the items accepted by $\mathsf{ORACLE}_\gamma^\star$, listed below. In each, we show that the value obtained is at least $(\gamma - \epsilon)V/2$.

- If the total weight of items packed by $\mathsf{ORACLE}_\gamma^\star(\mathcal{I})$ is strictly less than $(\gamma - \epsilon)$, then we know the total weight of items with value density $\geq d_\gamma^\star$ in the optimal solution's knapsack is strictly less than $(\gamma - \epsilon)$, and that the total weight of items with value density $\geq d_\gamma^\star$ in the instance is also strictly less than $(\gamma - \epsilon)$. By definition of $d_\gamma^\star$, $\mathsf{ORACLE}_\gamma^\star(\mathcal{I})$ obtains a value of $\geq \gamma V/2$.

- If the total weight of items packed by $\mathsf{ORACLE}_\gamma^\star(\mathcal{I})$ is $\geq (\gamma - \epsilon)$, then we know that $\gamma d_\gamma^\star \geq \gamma V/2$. If this wasn't true, there would exist some other value density $\hat{d}$ in $\mathsf{OPT}(\mathcal{I})$'s knapsack such that $\hat{d} > d_\gamma^\star$ and the total value of items with value density $\geq \hat{d}$ in $\mathsf{OPT}(\mathcal{I})$'s knapsack would have value at least $\gamma V/2$. Thus, $\mathsf{ORACLE}_\gamma^\star(\mathcal{I})$ obtains a value of at least $(\gamma - \epsilon)d_\gamma^\star \geq (\gamma - \epsilon)V/2$.

Therefore, $\mathsf{ORACLE}_\gamma^\star$ admits a value of at least $(\gamma - \epsilon)V/2$, and its approximation factor is at most $2/(\gamma - \epsilon)$.

**Lower bound:** Consider an input formed by a large number of infinitesimal items of density $L$ and total weight 1, followed by infinitesimal items of density $U$ and total weight $L/U$. An optimal algorithm accepts all items of density $U$ and fills the remaining space with items of density $L$, giving its knapsack a total value of $(L/U)U + (1 - L/U)L = 2L - L^2/U$. Any deterministic algorithm that satisfies 1-CTIF, however, must accept either all items of density $L$, giving its knapsack a value of $L$, or reject all items of density $L$, giving it a value of $(L/U)U = L$. In both cases, the approximation factor of the algorithm would be $\frac{2L - L^2/U}{L} = 2 - L/U$. □

We use $\mathsf{ORACLE}^\star_\gamma$ as a benchmark. Fix an arbitrary input $\mathcal{I} \in \Omega$. Let $\mathsf{LA\text{-}ECT}[\gamma]$ terminate filling $z_T$ fraction of the knapsack and obtaining a value of $\mathsf{LA\text{-}ECT}[\gamma](\mathcal{I})$.

Let $\mathsf{ORACLE}^\star_\gamma$ terminate obtaining a value of $\mathsf{ORACLE}^\star_\gamma(\mathcal{I})$.

Now we consider two cases – the case where $z_T < \kappa + \gamma$, and the case where $z_T \geq \kappa + \gamma$. We explore each below.

**Case I.** If $z_T < \kappa + \gamma$, the following statements must be true:

- Since the threshold function $\Psi^{\hat{d}_\gamma}(z) \leq d^\star_\gamma$ for all values $z$ less than $\kappa + \gamma$, any item accepted by $\mathsf{ORACLE}^\star_\gamma(\mathcal{I})$ must be accepted by $\mathsf{LA\text{-}ECT}[\gamma](\mathcal{I})$.

- Thus, $\mathsf{LA\text{-}ECT}[\gamma](\mathcal{I}) \geq \mathsf{ORACLE}^\star_\gamma(\mathcal{I})$, and $\frac{2}{\gamma - \epsilon}\mathsf{LA\text{-}ECT}[\gamma](\mathcal{I}) \geq \mathsf{OPT}(\mathcal{I})$.

Note that **Case I** implies that as $\gamma$ approaches 1, the value obtained by $\mathsf{LA\text{-}ECT}[1](\mathcal{I})$ is greater than or equal to that obtained by $\mathsf{ORACLE}^\star_1(\mathcal{I})$, and the competitive bound reduces to $\frac{\mathsf{OPT}(\mathcal{I})}{\mathsf{LA\text{-}ECT}[1](\mathcal{I})} \leq \frac{2}{1 - \epsilon}$.

**Case II.** If $z_T \geq \kappa + \gamma$, then we know that any item accepted by $\mathsf{ORACLE}^\star_\gamma(\mathcal{I})$ must have been accepted by $\mathsf{LA\text{-}ECT}[\gamma]$.

Proof by contradiction: assume that $z_T \geq \kappa + \gamma$ and there exists some item accepted by $\mathsf{ORACLE}^\star_\gamma(\mathcal{I})$ that wasn't accepted by $\mathsf{LA\text{-}ECT}[\gamma]$. This implies that when the item arrived to $\mathsf{LA\text{-}ECT}[\gamma]$, the threshold function $\Psi^{\hat{d}_\gamma}(z)$ was greater than $d^\star_\gamma$, which is the minimum acceptable value density for any item accepted by $\mathsf{ORACLE}^\star_\gamma(\mathcal{I})$.

Since $\Psi^{\hat{d}_\gamma}(z) \leq \hat{d}_\gamma$ for all values $z \leq \kappa + \gamma$ and $z_T \geq \kappa + \gamma$ implies that $\mathsf{LA\text{-}ECT}[\gamma]$ saw *enough* items with value density $\geq d^\star_\gamma$ to fill a $\gamma$ fraction of the knapsack, this causes a contradiction. Since items arrive in the same order to both $\mathsf{ORACLE}^\star_\gamma(\mathcal{I})$ and $\mathsf{LA\text{-}ECT}[\gamma]$, $\mathsf{ORACLE}^\star_\gamma(\mathcal{I})$'s knapsack would already be full by the time this item arrived.

This tells us that $\mathsf{LA\text{-}ECT}[\gamma](\mathcal{I}) \geq \mathsf{ORACLE}^\star_\gamma(\mathcal{I})$, and thus we have the following:

$$\frac{2}{\gamma - \epsilon}\mathsf{LA\text{-}ECT}[\gamma](\mathcal{I}) \geq \mathsf{OPT}(\mathcal{I})$$

It follows in either case that $\mathsf{LA\text{-}ECT}[\gamma]$ is $\frac{2}{\gamma}$-consistent for accurate predictions.

□

**Theorem 4.12.** *For any $\gamma \in [0, 1]$, and any $\mathcal{I} \in \Omega$, $\mathsf{LA\text{-}ECT}[\gamma]$ is $\frac{1}{1-\gamma}\left(\ln(U/L) + 1\right)$-robust.*

*Proof.* Fix an arbitrary input sequence $\mathcal{I}$. Let $\mathsf{LA\text{-}ECT}[\gamma]$ terminate filling $z_T$ fraction of the knapsack and obtaining a value of $\mathsf{LA\text{-}ECT}[\gamma](\mathcal{I})$. Let $\mathcal{P}$ and $\mathcal{P}^\star$ respectively be the sets of items picked by $\mathsf{LA\text{-}ECT}[\gamma]$ and the optimal solution.

Denote the weight and the value of the common items (items picked by both $\mathsf{LA\text{-}ECT}$ and $\mathsf{OPT}$) by $W = w(\mathcal{P} \cap \mathcal{P}^\star)$ and $V = v(\mathcal{P} \cap \mathcal{P}^\star)$. For each item $j$ which is *not* accepted by $\mathsf{LA\text{-}ECT}[\gamma]$, we know that its value density is $< \Psi^{\gamma, \hat{d}}(z_j) \leq \Psi^{\gamma, \hat{d}}(z_T)$ since $\Psi^{\gamma, \hat{d}}$ is a non-decreasing function of $z$. Thus, we have:

$$\mathsf{OPT}(\mathcal{I}) \leq V + \Psi^{\gamma, \hat{d}}(z_T)(1 - W).$$

Since $\text{LA-ECT}[\gamma](\mathcal{I}) = V + v(\mathcal{P} \setminus \mathcal{P}^\star)$, the inequality above implies that:

$$\frac{\text{OPT}(\mathcal{I})}{\text{LA-ECT}[\gamma](\mathcal{I})} \leq \frac{V + \Psi^{\gamma,\hat{d}}(z_T)(1 - W)}{V + v(\mathcal{P} \setminus \mathcal{P}^\star)}.$$

Note that each item $j$ picked in $\mathcal{P}$ must have value density of at least $\Psi^{\gamma,\hat{d}}(z_j)$, where $z_j$ is the knapsack utilization when that item arrives. Thus, we have that:

$$V \geq \sum_{j \in \mathcal{P} \cap \mathcal{P}^\star} \Psi^{\gamma,\hat{d}}(z_j)\, w_j =: V_1,$$

$$v(\mathcal{P} \setminus \mathcal{P}^\star) \geq \sum_{j \in \mathcal{P} \setminus \mathcal{P}^\star} \Psi^{\gamma,\hat{d}}(z_j)\, w_j =: V_2.$$

Since $\text{OPT}(\mathcal{I}) \geq \text{LA-ECT}[\gamma](\mathcal{I})$, we have that

$$\frac{\text{OPT}(\mathcal{I})}{\text{LA-ECT}[\gamma](\mathcal{I})} \leq \frac{V + \Psi^{\gamma,\hat{d}}(z_T)(1 - W)}{V + v(\mathcal{P} \setminus \mathcal{P}^\star)} \leq \frac{V_1 + \Psi^{\gamma,\hat{d}}(z_T)(1 - W)}{V_1 + V_2}.$$

Note that $V_1 \leq \Psi^{\gamma,\hat{d}}(z_T) w(\mathcal{P} \cap \mathcal{P}^\star) = \Psi^{\gamma,\hat{d}}(z_T) W$, and so, plugging in the actual values of $V_1$ and $V_2$, we get:

$$\frac{\text{OPT}(\mathcal{I})}{\text{LA-ECT}[\gamma](\mathcal{I})} \leq \frac{\Psi^{\gamma,\hat{d}}(z_T)}{\sum_{j \in \mathcal{P}} \Psi^{\gamma,\hat{d}}(z_j)\, w_j}.$$

Based on the assumption that individual item weights are much smaller than 1, we can substitute for $w_j$ with $\delta z_j = z_{j+1} - z_j$ for all $j$. This substitution allows us to obtain an approximate value of the summation via integration:

$$\sum_{j \in \mathcal{P}} \Psi^{\gamma,\hat{d}}(z_j)\, w_j \approx \int_0^{z_T} \Psi^{\gamma,\hat{d}}(z) dz$$

Now we consider three separate cases – the case where $z_T \in [0, \kappa)$, the case where $z_T \in [\kappa, \kappa + \gamma)$, and the case where $z_T \in [\kappa + \gamma, 1]$. We explore each below.

**Case I.** If $z_T \in [0, \kappa)$, $\text{OPT}(\mathcal{I})$ is bounded by $\Psi^{\gamma,\hat{d}}(z_T) \leq \hat{d}$. Furthermore,

$$\sum_{j \in \mathcal{P}} \Psi^{\gamma,\hat{d}}(z_j)\, w_j \approx \int_0^{z_T} \Psi^{\gamma,\hat{d}}(z) dz = \int_0^{z_T} \left(\frac{Ue}{L}\right)^{\frac{z}{1-\gamma}} \left(\frac{L}{e}\right) dz$$

$$= (1 - \gamma)\left(\frac{L}{e}\right)\left[\frac{\left(\frac{Ue}{L}\right)^{\frac{z}{1-\gamma}}}{\ln(Ue/L)}\right]_0^{z_T} = (1 - \gamma)\frac{\Psi^{\gamma,\hat{d}}(z_T)}{\ln(Ue/L)}$$

Combined with the previous equation for the competitive ratio, this gives us the following:

$$\frac{\text{OPT}(\mathcal{I})}{\text{LA-ECT}[\gamma](\mathcal{I})} \leq \frac{\Psi^{\gamma,\hat{d}}(z_T)}{(1 - \gamma)\frac{\Psi^{\gamma,\hat{d}}(z_T)}{\ln(U/L)+1}} \leq \frac{\hat{d}}{(1 - \gamma)\frac{\hat{d}}{\ln(U/L)+1}} = \frac{1}{1 - \gamma}\left(\ln(U/L) + 1\right).$$

**Case II.** If $z_T \in [\kappa, \kappa + \gamma)$, $\text{OPT}(\mathcal{I})$ is bounded by $\Psi^{\gamma,\hat{d}}(z_T) = \hat{d}$. Furthermore,

$$\int_0^{z_T} \Psi^{\gamma,\hat{d}}(z) dz = \int_0^{\kappa} \left(\frac{Ue}{L}\right)^{\frac{z}{1-\gamma}}\left(\frac{L}{e}\right) dz + \int_\kappa^{z_T} \hat{d}\, dz \geq \int_0^\kappa \left(\frac{Ue}{L}\right)^{\frac{z}{1-\gamma}}\left(\frac{L}{e}\right) dz.$$

Note that since the bound on $\text{OPT}(\mathcal{I})$ can be the same (i.e. $\text{OPT}(\mathcal{I}) \leq \hat{d}$), Case I is strictly worse than Case II for the competitive ratio, and we inherit the worse bound:

$$\frac{\text{OPT}(\mathcal{I})}{\text{LA-ECT}[\gamma](\mathcal{I})} \leq \frac{\hat{d}}{(1 - \gamma)\frac{\hat{d}}{\ln(U/L)+1}} = \frac{1}{1 - \gamma}\left(\ln(U/L) + 1\right).$$

**Case III.** If $z_T \in [\kappa + \gamma, 1]$, $\mathsf{OPT}(\mathcal{I})$ is bounded by $\Psi^{\gamma,\hat{d}}(z_T) \le U$. Furthermore,

$$\sum_{j \in \mathcal{P}} \Psi^{\gamma,\hat{d}}(z_j)\, w_j \approx \int_0^{z_T} \Psi^{\gamma,\hat{d}}(z)dz = \int_0^{z_T-\gamma} \left(\frac{Ue}{L}\right)^{\frac{z}{1-\gamma}}\left(\frac{L}{e}\right) dz + \int_\kappa^{\kappa+\gamma} \hat{d}\, dz$$

$$= (1-\gamma)\left(\frac{L}{e}\right)\left[\frac{\left(\frac{Ue}{L}\right)^{\frac{z}{1-\gamma}}}{\ln(Ue/L)}\right]_0^{z_T-\gamma} + \gamma\hat{d}$$

$$= (1-\gamma)\frac{\Psi^{\gamma,\hat{d}}(z_T)}{\ln(Ue/L)} + \gamma\hat{d}.$$

Combined with the previous equation for the competitive ratio, this gives us the following:

$$\frac{\mathsf{OPT}(\mathcal{I})}{\mathsf{LA\text{-}ECT}[\gamma](\mathcal{I})} \le \frac{\Psi^{\gamma,\hat{d}}(z_T)}{(1-\gamma)\frac{\Psi^{\gamma,\hat{d}}(z_T)}{\ln(U/L)+1} + \gamma\hat{d}} \le \frac{\Psi^{\gamma,\hat{d}}(z_T)}{(1-\gamma)\frac{\Psi^{\gamma,\hat{d}}(z_T)}{\ln(U/L)+1}} = \frac{1}{1-\gamma}\left(\ln(U/L) + 1\right).$$

Since we have shown that $\mathsf{LA\text{-}ECT}[\gamma](\mathcal{I})$ obtains at least $\frac{1}{1-\gamma}\left(\ln(U/L) + 1\right)$ of the value obtained by $\mathsf{OPT}(\mathcal{I})$ in each case, we conclude that $\mathsf{LA\text{-}ECT}[\gamma](\mathcal{I})$ is $\frac{1}{1-\gamma}\left(\ln(U/L) + 1\right)$-robust. $\square$

**Theorem 4.13.** *For any learning-augmented online algorithm* $\mathsf{ALG}$ *which satisfies* 1-CTIF*, one of the following holds: Either* $\mathsf{ALG}$*'s consistency is* $> 2\sqrt{U/L} - 1$*, or* $\mathsf{ALG}$ *has unbounded robustness. Furthermore, the consistency of any algorithm is lower bounded by* $2 - \varepsilon^2/1+\varepsilon$*, where* $\varepsilon = \sqrt{L/U}$*.*

*Proof.* We begin by proving the first statement, which gives a *consistency-robustness* trade off for any learning-augmented $\mathsf{ALG}$.

**Lemma C.3.** *One of the following statements holds for any* 1-CTIF *online algorithm* $\mathsf{ALG}$ *with any prediction:*

*(i)* $\mathsf{ALG}$ *has consistency worse (larger) than* $2\sqrt{U/L} - 1$*.*

*(ii)* $\mathsf{ALG}$ *has unbounded robustness.*

**Proof of Lemma C.3.** Let $\varepsilon = \sqrt{L/U}$ and $V = \sqrt{LU}$ and note that $\varepsilon V = L$ and $V/\varepsilon = U$.

Consider a sequence $\mathcal{I}$ that starts with a set of *red items* of density $L$ and total size 1, continues with $1/\varepsilon$ "white" items, each of size $\varepsilon$ and density $\varepsilon(1+\varepsilon)V$ (which is in $[L, U]$), and ends with one *black* item of size $\varepsilon$ and density $V/\varepsilon = U$. The optimal solution rejects all red items and accepts all other items except one white item.
The optimal profit is thus $\mathsf{OPT}(\mathcal{I}) = (1+\varepsilon)V - \varepsilon(1+\varepsilon)V + V = (2-\varepsilon^2)V$.

Suppose the predictions are consistent with $\mathcal{I}$. Then, a 1-CTIF learning-augmented $\mathsf{ALG}$ has the following two choices:

- It accepts all red items. Then if the input is indeed $\mathcal{I}$, the consistency of $\mathsf{ALG}$ would be $\frac{(2-\varepsilon^2)\sqrt{LU}}{L} = (2-\varepsilon^2)\sqrt{U/L} = 2\sqrt{U/L} - \sqrt{L/U} > 2\sqrt{U/L} - 1$. In this case, **(i)** holds.

- It rejects all red items. Then, the input may be formed entirely by the red items (and the predictions are incorrect). The algorithm does not accept any item, and its robustness will be unbounded. In this case, **(ii)** holds. $\square$

Note that $\mathsf{LA\text{-}ECT}[\gamma]$ satisfies **(ii)** when $\gamma \to 1$.

Next, we prove the final statement, which lower bounds the achievable consistency for any 1-CTIF algorithm. To do this, we consider a semi-online 1-CTIF algorithm $\mathsf{ALG}$. It has full knowledge of the items in the instance, but must process items sequentially using a threshold it has to set in advance. Items still arrive in an online manner, decisions are immediate and irrevocable, and the order of arrival is unknown.

**Lemma C.4.** *Any semi-online* 1-CTIF *algorithm has an approximation factor of at least* $\frac{2-\varepsilon^2}{1+\varepsilon}$, *where* $\varepsilon = \sqrt{L/U}$.

**Proof of Lemma C.4.** As previously, let $V = \sqrt{LU}$ and note that $\varepsilon V = L$ and $V/\varepsilon = U$.

Consider an input sequence starting with $1/\varepsilon$ "white" items, each of size $\varepsilon$ and density $\varepsilon(1+\varepsilon)V$ (which is in $[L, U]$). Note that white items have a total size of 1 (knapsack capacity) and a total value of $(1+\varepsilon)V$. Suppose the input continues with one *black* item of size $\varepsilon$ and density $V/\varepsilon = U$. An optimal algorithm accepts all items in the input sequence except one white item. The optimal profit is thus $(1+\varepsilon)V - \varepsilon(1+\varepsilon)V + V = (2 - \varepsilon^2)V$.

Given that the entire set of white items fits in the knapsack, any 1-CTIF algorithm must accept or reject them all. In the former case, the algorithm cannot accept the black item (the knapsack becomes full before processing the black item), and its profit will be $(1+\varepsilon)V$, resulting in an approximation factor of $\frac{2-\varepsilon^2}{1+\varepsilon}$. In the latter case, the algorithm can only accept the black item, and its approximation factor would be at least $2 - \varepsilon^2$.

Combining the statements of Lemmas C.3 and C.4, the original statement follows. □

