# OpenReview forum: "Time Fairness in Online Knapsack Problems"
_ICLR.cc/2024/Conference — ICLR 2024 poster_

### Official Review · Reviewer_rQKw · 2023-10-21

**Soundness:** 3 good
**Presentation:** 3 good
**Contribution:** 3 good
**Rating:** 6
**Confidence:** 3

**Summary:**

The authors study time fairness in online knapsack problems (OKPs). Since time-independent fairness is not achievable in OKP, they study the theoretical trade off between competitiveness and fairness with the notion of alpha-conditional time-independent fairness (alpha-CTIF). Several deterministic and randomized algorithms (including some proposed by the authors) are discussed. Numerical experiments comparing these algorithms are conducted on three OKP instances.

**Strengths:**

The paper is well written and the results are technically sound. The notion of alpha-CTIF is new. In the deterministic case, both a lower bound on the achievable competitive ratio and an alpha-CTIF online deterministic algorithm achieving this lower bound are given.

**Weaknesses:**

1. The definition of alpha-CTIF does not come naturally to me. Since this definition is new, I would appreciate more motivation behind its use.
2. Numerical experiments are conducted on a limited number of instances, and therefore not very informative.

**Questions:**

The authors mention that the notion of alpha-CTIF can be potentially adaptable to other online problems. How should one genearlize this notion to other problems? Can they give some examples?

---

> ### Author Response · Authors · 2023-11-15
>
> Thanks for your review and comments about our paper! Please find our response below:
>
> - **(motivation for α-CTIF)**
> We will carefully revise the presentation to clarify the definition based on feedback.
> In Example 1.1, we describe an empirical problem of cloud resource management, which reduces to the online knapsack problem.
> Our study of the tradeoff between static and dynamic pricing (theoretically, α-CTIF) is primarily motivated by the observation that cloud resource providers almost never use dynamic pricing in the real world because it is inconvenient for and *inequitable towards* consumers.  Since the canonical optimal ZCL algorithm corresponds to a nearly fully dynamic pricing strategy, there is a disconnect between the theoretically optimal revenue maximization strategy and the static pricing which is actually used.
> Static pricing is simple and interpretable, but has drawbacks in terms of revenue maximization and demand response.  Our work then considers how we can interpolate between these two extremes in a theoretically rigorous way, which corresponds to a formalization of the “surge pricing” model employed by e.g. ridesharing platforms.  This motivates the definition of α-CTIF, where α allows us to tune the tradeoff between these two pricing models.
>
>
> - **(adapting fairness to other online problems)**
> We do believe that the notion of α-CTIF can be adapted easily for other online problems, and will expand on this potential generalization.  We think the notion is flexible and natural in many other contexts, and our hope is that this work is seen to be as much about the general framework and techniques as it is about the specific OKP setting.
> In general, α-CTIF should be applicable to any online problem where decisions are of an “accept or reject” nature.  A special case of α-CTIF has already been studied in [1] for prophet inequalities, and further generalizations may apply to online problems such as time-series search [2], interval scheduling [3], independent set [4], bipartite matching [5] or multiple knapsack [6].
> In each of these online problems, an irrevocable (and possibly weighted) accept/reject decision must be made at each time step. The high-level notion of time fairness then captures the concept that items should be treated fairly in the online decision-making process, regardless of their arrival position in the sequence.
> For instance, α-CTIF could be meaningful in a problem such as online bipartite matching [5], where the nodes in one “half” of a bipartite graph arrive online and the algorithm must make an irrevocable matching to the offline nodes, which may be weighted (e.g. of different quality). Most traditionally competitive approaches to this problem would tend to match early arrival nodes to higher-quality offline nodes, while matching late arrivals with lower-quality offline nodes, thereby violating an equitable notion of access across arrival times, and motivating our definitions of time fairness across nodes arriving at different times.
> We believe that time fairness is a challenging, intellectually interesting, and yet empirically important constraint to consider in the context of many such online problems.
>
> [1] M. Arsenis and R. Kleinberg. *Individual fairness in prophet inequalities*. EC ’22. doi:10.1145/3490486.3538301.
>
> [2] R. El-Yaniv, A. Fiat, R. M. Karp, and G. Turpin. *Optimal search and one-way trading online algorithms*.
> Algorithmica, 30(1):101–139, May 2001. doi:10.1007/s00453-001-0003-0
>
> [3] R. Lipton and A. Tomkins. *Online Interval Scheduling*. SODA ‘94. doi:10.5555/314464.314506
>
> [4] M. M. Halldórsson, K. Iwama, S. Miyazaki, and S. Taketomi. *Online independent sets*. Theoretical Computer Science, (289)2:953-962, 2002. doi:10.1016/S0304-3975(01)00411-X.
>
> [5] R. M. Karp, U. V. Vazirani, and V. V. Vazirani. *An optimal algorithm for on-line bipartite matching*. STOC ‘90.
>
> [6] M. Cygan, Ł. Jeż, and J. Sgall. *Online knapsack revisited*. Theory of Computing Systems 58 (2016): 153-190.

---

> > ### Comment · Reviewer_rQKw · 2023-11-15
> >
> > I appreciate the authors' detailed reply. Could you please explain to me why the fairness notion in [1] is a special case of $\alpha$-CTIF?

---

> > > ### Author Response · Authors · 2023-11-15
> > >
> > > Certainly!  We will be sure to clarify this in the presentation of our definition as well.
> > >
> > > In [1], the authors study a notion of time-independent fairness (TIF) in the context of prophet inequalities.  Prophet inequalities give competitive guarantees for online algorithms for the classic “secretary problem”, where a decision maker sequentially evaluates candidates (whose “quality” can be represented by, e.g., a real value) and must decide whether to hire the candidate without seeing future candidates.
> > >
> > > While the secretary problem under prophet inequalities and OKP have different underlying assumptions (i.e., prophet inequalities generally assume stochastic and/or random order inputs, while the OKP variant we study is fully adversarial), they are similar in the sense that they both make irrevocable admission decisions online.  As such, we can construct an illustrative OKP sequence, which can be seen to generalize the secretary problem as follows: set the knapsack capacity (corresponding to the open position) to be 1, and the items (corresponding to the candidates) to have weight one each.  Then, the differences between items are exactly the values, as in the secretary problem.
> > >
> > > To recover the notion of TIF studied in [1, Def. 3.2] from our definition of α-CTIF, we can set $\alpha = 1$ in our main definition (Definition 3.6) and consider a sequence such as the illustrative one described above.
> > > The only remaining difference, namely, the “conditional” portion of the definition, can be attributed to a slight difference in the problem setting.  Put simply, in the secretary problem, as soon as the player hires a candidate, the process terminates.  In contrast, in OKP, once the knapsack is full, the process does not terminate; instead, the player is forced to reject any items that will not fit into the knapsack.  This motivates the inclusion of a conditional fairness constraint, which avoids the impossibility results discussed in Theorem 3.3 and the beginning of Section 3.2 -- when our definition of α-CTIF is applied to the secretary problem, the condition is redundant.

---

> > > > ### Comment · Reviewer_rQKw · 2023-11-16
> > > >
> > > > Thanks for the clarification. I intend to keep my score.

---

### Official Review · Reviewer_zsDR · 2023-10-30

**Soundness:** 3 good
**Presentation:** 3 good
**Contribution:** 3 good
**Rating:** 6
**Confidence:** 4

**Summary:**

This paper considers a variant of the online knapsack problem in which items arrive online, and the algorithm does not know the weight and value of the item before it arrives. Upon each arrival, the algorithm must make an irrevocable decision to either accept or reject the item. The goal is to accept a set of items such that (1) the total weight of the accepted item does not exceed the capacity of the knapsack; (2) the total value of the accepted items is maximized. By observing that the algorithm for the online knapsack problem may not be fair for each item, the authors include a fairness constraint to the problem, which leads to a new variant of the problem.
The main contribution is a fairness notation, and they show that such a fairness notation is too strong to admit a bounded competitive algorithm. They then relax the constraint and propose a deterministic algorithm that is Pareto optimal on fairness and efficiency. In addition, the authors also consider the learning-augmented algorithms. This work also includes some numerical experiments.

**Strengths:**

1.The studied problem is generally well-motivated, i.e., it aims to include the fairness constraint to the online knapsack problem. The motivation is clear, although I think the example given in the paper is not convincing. See the weakness below.
2.The paper is solidly technical. Although the basic idea of the algorithm comes from the classical threshold algorithm of OKP, the proposed algorithm contains sufficient new ideas and analysis.
3.The results included in the paper are rich. The authors provide a lower bound for the studied problem. In addition, the authors show that randomness can further improve the ratio. Furthermore, the learning-augmented algorithm is also considered in the paper.

**Weaknesses:**

1.The main weakness is that the motivation of the definition of \alpha-CTIF is not clear. I agree that \alpha-CTIF is a well-defined definition. I have taken a quick look at the proof of Theorem 4.3, and I also understand that such a definition plays an important role in the analysis. But the practical meaning of such a relaxation is not clear. For example, when \alpha=0 or epsilon, what does \alpha-CTIF stand for? This part is important since the main purpose of the work is to make the OKP’s algorithm fair. Thus, finding a suitable fairness notion is critical.
2.It seems that the authors try to claim that fairness is important in the online knapsack problem by Example 1.1. But Example 1.1 is just a typical example of the online knapsack problem, and it does not reflect the importance of fairness. One may need to find another example to convince people about the importance of fairness.

**Questions:**

Please see weakness above.

---

> ### Author Response · Authors · 2023-11-15
>
> Thank you for your review and comments about our paper! Please find our response below:
>
> - **(definition of α-CTIF (1))**
> We will clarify the motivation behind the definition based on feedback.
> The practical meaning of the α-CTIF constraint is simply that we maintain our desired objective of time fairness in admitting items over at least an α-fraction of the knapsack capacity.  More formally, during the entirety of this α-fair region, an item’s probability of admission is independent of its arrival time within the input sequence, *as long as* there is capacity for it. This last technical capacity condition turns out to be necessary to get absolutely any fairness guarantees whatsoever (Theorem 3.3).
> Since our motivation is to study the inherent quantitative tradeoff between the competitive ratio (performance) and fairness, we use the parameter α in the notion of α-CTIF to precisely quantify the amount of fairness: larger α corresponds to a stronger fairness constraint.
> We note that the definition of α-CTIF is not meaningful for $\alpha < 1/ \ln(U/L)+1$ (including $\alpha = 0$ or $\alpha = \epsilon$). For any value of $\alpha \leq 1/ \ln(U/L)+1$, α-CTIF is already attained by the canonical online ZCL algorithm [1], which is optimally competitive. We discuss this point in the proof of Proposition 4.2 (in the appendix), but it is not stated explicitly in the main body of the paper – we will carefully revise and explicitly mention this observation where it is appropriate.
> In particular, α-CTIF is only meaningful and of interest when $\alpha > 1/ \ln(U/L)+1$. In practical terms, this corresponds to the case where the static pricing period is longer, compared to the dynamic but “unfair” pricing approach used by the exponential threshold function of the optimal ZCL algorithm.
>
> - **(importance of fairness (2))**
> We will carefully revise the presentation to clarify the following motivation.
> It is true, as you point out, that Example 1.1 is a typical example of the online knapsack problem, but our study of the tradeoff between competitiveness and time fairness (corresponding broadly to the tradeoff between dynamic and static pricing) is motivated by the observation that cloud resource providers almost never use dynamic pricing in the real world, because it is inconvenient for and *inequitable towards* consumers.
> Theoretical studies [2] have shown that dynamic pricing strategies (such as the ZCL algorithm) for the cloud resource management problem can improve performance.  However, these dynamic pricing strategies introduce price discrimination; clients may need to be charged different prices for the same resource to achieve the best performance bound. The long-term market impacts of such price discrimination are unclear, although dynamic pricing has seen success in airline ticket pricing and ride-sharing pricing applications.
> In cloud resource management, the dominant paradigm used in practice is static pricing, which is simple and interpretable from the consumer perspective (and roughly corresponds to increased values of α in α-CTIF), but comes with drawbacks in terms of revenue maximization and demand response.
> Thus, we aim to study the fairness issue based on this example, proposing α-CTIF as a theoretically rigorous method to capture the trade-off between static vs. dynamic pricing (resp. fairness vs. competitive ratio), where the value of α allows us to interpolate between these two opposing pricing models.  Using such a combination of static and dynamic pricing can be interpreted as a formalization of the “surge pricing” model used by, e.g., rideshare platforms [3].  Beyond cloud resource management, we also anticipate that this tradeoff between static and dynamic pricing could be broadly applicable.
>
> [1] Y. Zhou, D. Chakrabarty, and R. Lukose. *Budget constrained bidding in keyword auctions and online knapsack problems*. WWW ’08. doi:10.1145/1367497.1367747
>
> [2] Z. Zhang, Z. Li, and C. Wu. *Optimal posted prices for online cloud resource allocation*. SIGMETRICS Perform. Eval. Rev., 45(1):60, 2017. doi:10.1145/3143314.3078529.
>
> [3] J. Hall, C. Kendrick, and C. Nosko. *The effects of Uber’s surge pricing: A case study*. The University of Chicago Booth School of Business, 2015.

---

> > ### Comment · Reviewer_zsDR · 2023-11-16
> > **Thank you!**
> >
> > Thank you for addressing my questions. I have read the response. Now, I got the relaxation of alpha-CTIF. It basically relaxes the probability of acceptance probability of an item so that the arrival order also has a slight impact on the acceptance probability. I think the relaxation makes sense, and I have improved my score.

---

> > > ### Author Response · Authors · 2023-11-16
> > >
> > > We appreciate your comments and thank you for the improved score.

---

### Official Review · Reviewer_37F8 · 2023-11-01

**Soundness:** 3 good
**Presentation:** 4 excellent
**Contribution:** 3 good
**Rating:** 8
**Confidence:** 2

**Summary:**

This paper studies the online knapsack problem (OKP) from the perspective of fairness. In particular, the authors are interested in competitive algorithms that don’t discriminate items based on when they appear in the sequence of observed items. First, they formalize the notion of time fairness. Second, they observe that previous OKP algorithms with good competitive ratios are not time-fair. Third, they show that randomization can be used to reach Pareto-optimal trade-off between fairness and competitiveness. Pareto optimality here, means that neither measure can be increased without decreasing the other. Fourth, they run empirical experiments, which show poor performance of the randomized algorithm. The authors liken this to the results of Reineke in 2014, where similarly poor empirical performance was observed on theoretically strong randomized algorithms for caching.  Fifth, they show that learning-augmented algorithms can be used to improve both fairness and competitiveness. Finally, they show the superior experimental performance of the learning-augmented algorithms.

**Strengths:**

Originality: Online knapsack is well studied, but this paper is the first to study the fairness aspect of the problem under adversarial sequences.

Result Quality: The investigation is thorough. The authors combine ideas from mathematics, economics, and experimentation effectively to produce nice theoretical contributions that are driven farther and validated by experimentation.

Writing Quality / Clarity: I am a non-expert in this field. I found the paper well-written. I was especially pleased with the clear introduction.

Significance: The result is a good quality contribution for ICLR.

**Weaknesses:**

Originality:  If I understand correctly, there are previous papers that study fairness in OKP when the sequence is not adversarial. Not necessarily a weakness of this paper, but an observation that is relevant about the originality of the study.

Significance: Like most papers at ICLR, this paper is probably of interest to a small subcommunity at the conference. I believe this is true of most papers at the conference, so it is not a negative comment towards a paper.

**Questions:**

Are there any Pareto-optimality claims to be made about the learning-augmented algorithms that you describe in the later part of the paper?

---

> ### Author Response · Authors · 2023-11-15
>
> Thanks for your review and positive comments about our paper!  Please find our response below:
> - **(Pareto-optimality of the proposed learning-augmented algorithm)**
> We will add a remark in the paper to clarify and highlight the following point:  in Theorem 4.13, we prove that our algorithm is on the Pareto frontier – namely, when $\gamma \to 1$, LA-ECT obtains the optimal consistency-robustness tradeoff.  It would be very interesting to study whether our algorithm achieves the Pareto-optimal trade-off for arbitrary values of $\gamma$. Thank you very much for the suggestion!

---

> > ### Comment · Reviewer_37F8 · 2023-11-20
> >
> > Thank you for your response. I would recommend clarifying this point in the paper.

---

### Meta-Review · Area_Chair_yDkc · 2023-12-04

**Metareview:**

This paper presents a new variant of the online knapsack problem.  The paper defines a fair variation and gives algorithms with bounded competitiveness for this variation. The reviewers found the new formulation of the problem interesting and well-motivated.  The algorithmic insights will be of interest to a subset of the ICLR community.

**Justification For Why Not Higher Score:**

While I believe the new problem is interesting, I think its scope is limited to a small subset of the ICLR community.  Due to this, I will only recommend acceptance.

**Justification For Why Not Lower Score:**

The new model and algorithm are interesting.  I see no good reason to delay publication of the results.

---

### Decision · Program_Chairs · 2024-01-16

Accept (poster)